# Adaptive Defense against Harmful Fine-Tuning for Large Language Models via Bayesian Data Scheduler

**Zixuan Hu**[1]    **Li Shen**[2]    **Zhenyi Wang**[3]    **Yongxian Wei**[4]    **Dacheng Tao**[1*]

[1]Nanyang Technological University    [2]Shenzhen Campus of Sun Yat-sen University

[3] University of Central Florida    [4] Tsinghua University

ZIXUAN014@e.ntu.edu.sg    mathshenli@gmail.com    zhenyi.wang@ucf.edu

weiyx23@mails.tsinghua.edu.cn    dacheng.tao@gmail.com

## Abstract

Harmful fine-tuning poses critical safety risks to fine-tuning-as-a-service for large language models. Existing defense strategies *preemptively* build robustness via attack simulation but suffer from fundamental limitations: (i) the infeasibility of extending attack simulations beyond bounded threat models due to the inherent difficulty of anticipating unknown attacks, and (ii) limited adaptability to varying attack settings, as simulation fails to capture their variability and complexity. To address these challenges, we propose Bayesian Data Scheduler (BDS), an *adaptive* tuning-stage defense strategy with no need for attack simulation. BDS formulates harmful fine-tuning defense as a Bayesian inference problem, learning the posterior distribution of each data point's safety attribute, conditioned on the fine-tuning and alignment datasets. The fine-tuning process is then constrained by weighting data with their safety attributes sampled from the posterior, thus mitigating the influence of harmful data. By leveraging the post hoc nature of Bayesian inference, the posterior is conditioned on the fine-tuning dataset, enabling BDS to tailor its defense to the specific dataset, thereby achieving adaptive defense. Furthermore, we introduce a neural scheduler based on amortized Bayesian learning, enabling efficient transfer to new data without retraining. Comprehensive results across diverse attack and defense settings demonstrate the state-of-the-art performance of our approach. Code is available at https://github.com/Egg-Hu/Bayesian-Data-Scheduler.

## 1 Introduction

Fine-tuning-as-a-service has become a widely adopted paradigm among mainstream LLM providers (*e.g.*, OpenAI[2]), enabling the delivery of customized language model solutions. In this paradigm, users upload demonstration data representing their desired behaviors, and providers fine-tune the foundational models on their behalf. Despite its growing adoption, recent red teaming studies have uncovered a critical vulnerability: harmful fine-tuning [48, 29]. Harmful fine-tuning refers to that the presence of even a small fraction of harmful data in user-provided datasets can cause fine-tuned models to deviate from the safety alignment established during pre-training. This vulnerability introduces a substantial attack surface and also undermines the reliability and quality of the service.

Existing defenses [29, 33, 52] primarily rely on a *preemptive* manner, aiming to build robustness through attack simulation prior to fine-tuning. However, these defenses suffer from fundamental limitations: (i) the infeasibility of extending attack simulations beyond bounded threat models due to the inherent difficulty of anticipating unknown attacks; and (ii) limited adaptability to varying attack

---

[*]Corresponding author: Dacheng Tao

[2]Fine-tuning API by OpenAI: https://platform.openai.com/docs/guides/fine-tuning.

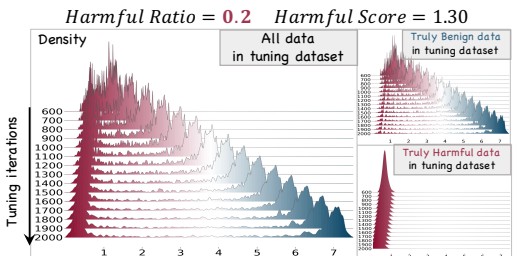 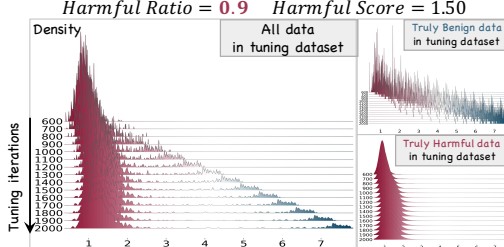

Inferred Data Weight (Higher Indicates Greater Likelihood of Being Benign Data)

Figure 1: For encountered datasets with unknown and different harmful ratios, BDS *adaptively* schedules data into higher and lower weight groups during tuning (*largest panels*). To verify correctness of our data scheduling, we observe that most truly benign data indeed receive higher weights (*top right panels*), while almost all truly harmful data consistently receive lower weights (*bottom right panels*).

settings, as the preemptive attack simulation fails to capture the variability and complexity of post hoc attacks. Consequently, their performance deteriorates sharply as the harmful data ratio increases or the attack strategy varies, rendering them ineffective against unpredictable attack surfaces.

To address these challenges, we propose Bayesian Data Scheduler (BDS), an adaptive fine-tuning-stage defense strategy with no need for attack simulation. BDS formulates harmful fine-tuning defense as a Bayesian inference problem, learning the posterior distribution of each data point's safety attribute, conditioned on the fine-tuning and alignment datasets. The fine-tuning is then constrained by weighting data with the safety attributes [3] sampled from the posterior, thus mitigating the negative influence of harmful data. Due to the post hoc nature of Bayesian inference, the posterior is conditioned on the encountered fine-tuning dataset, which enables BDS to precisely tailor its defense to the specific dataset, thus achieving adaptive defense. Additionally, as the posterior of safety attribute is datapoint-wise, *i.e.*, $p(w_i \mid z_i^{\text{ft}})$, independently learning distribution for each data point scales with the dataset size and requires relearning for new data. To address scalability and transferability issues, we further introduce a neural scheduler based on amortized Bayesian learning [51], enabling efficient transfer to new data without retraining.

We conduct comprehensive experiments on five diverse downstream datasets using three representative LLM architectures under a wide range of attack and defense settings. Results show that our approach achieves state-of-the-art (SOTA) performance, with a remarkable 74.4% improvement at a high harmful ratio of 0.9 and and an average boost of over 50% across ratios from 0 to 1. Moreover, BDS consistently maintains a remarkably low harmfulness score (around 1) across a wide range of advanced attack dynamics, including benign [48], out-of-distribution (OOD), and identity-shifting [48] attacks, demonstrating superior effectiveness, adaptiveness, and robustness.

To the end, we summarize our contributions as follows:

- For the first time, we formulate harmful fine-tuning defense as a Bayesian inference problem, offering a principled framework BDS achieving adaptive defense with no need for attack simulation.

- We propose two BDS implementations, with the neural version enabling efficient transfer to new data without retraining. Leveraging post hoc nature of Bayesian inference, BDS learns the posterior distribution of data weights conditioned on specific datasets, thus achieving adaptive defense.

- Comprehensive experiments on diverse attack and defense settings demonstrate the SOTA performance and superior adaptiveness of our proposed method.

## 2 Problem Definition

**Scenario.** The scenario considers users uploading a personalized fine-tuning dataset $\mathcal{D}_{\text{ft}}$, which the service provider uses to fine-tune their model $\theta$. Once fine-tuned, the personalized API is returned for user-specific applications.

**Threat model.** Harmful fine-tuning arises when an attacker deliberately uploads a fine-tuning dataset $\mathcal{D}_{\text{ft}}$ that contains a mix of benign and harmful data. Specifically, the dataset $\mathcal{D}_{\text{ft}}$ consists

---

[3] In this paper, "safety attribute" and "data weight" are used synonymously.

of a proportion $p$ of harmful data and $1 - p$ of benign data, such that $\mathcal{D}_{\text{ft}} = \mathcal{D}_{\text{ft}}^{\text{benign}} \cup \mathcal{D}_{\text{ft}}^{\text{harmful}}$. The attacker, acting as a user, has the capacity to control the composition of the fine-tuning dataset, including the proportion and content of harmful data, without requiring knowledge of the model's architecture or parameters. On the other hand, the defender, acting as the service provider, has the capacity to control the fine-tuning process.

**Assumption.** We assume that the service provider maintains an alignment dataset, denoted as $\mathcal{D}_{\text{safe}}$ (harmful prompt-safe answer pairs). The availability of such an alignment dataset has been previously discussed in previous works [52, 57] and it is available in publicly available resources (*e.g.*, BeaverTail [35]). While prior studies require an additional harmful dataset for attack simulation, constructing such a dataset is difficult due to lacking prior knowledge about unknown and varying attack data $\mathcal{D}_{\text{ft}}^{\text{harmful}}$. Our work overcomes this limitation with no need for a simulated harmful dataset.

# 3 Related work.

**Mechanism study of harmful fine-tuning.** Recent works have investigated why LLMs are highly sensitive to harmful fine-tuning. (i) Adversarial perturbations: Harmful fine-tuning can introduce adversarial shifts in model parameters [29, 46] or embeddings [33], leading to degraded safety alignment. (ii) Structural vulnerabilities: Other works highlight that certain layers or modules are more critical for maintaining safety than others [21, 37, 47]. (iii) *Catastrophic forgetting:* Safety alignment degradation has also been linked to *catastrophic forgetting* [43], where alignment knowledge tends to be lost under the sequential training paradigm [6, 13].

**Harmful fine-tuning attack.** On the attack side, [48] demonstrate that even fine-tuning solely on benign data can inadvertently weaken a model's safety alignment. Subsequent work [20] further identifies specific subsets of benign data that are particularly prone to degrading model safety after fine-tuning. [18] introduce covert malicious fine-tuning, which enhances the stealthiness of such attacks, while [32] construct harmful data designed to deliberately circumvent moderation guardrails.

**Harmful fine-tuning defense.** On the defense side, existing defense methods can be categorized into three main categories, including alignment stage [33, 52, 58, 42, 29, 40, 41, 7, 75], fine-tuning stage [44, 3, 31, 61, 79, 55], and post-fine-tuning stage solutions [76, 12, 28, 70, 62]. The method proposed in this paper should be classified into a fine-tuning stage solution. However, it differs fundamentally from prior work in that we propose a loss-based data scheduling mechanism derived from Bayesian inference principles, whereas previous approaches typically perform hard-label data selection based on less effective heuristic rules [15, 5] or manually tuned thresholds [8, 55]. Moreover, unlike the current SOTA method Booster [29], our method does not require attack simulation, making it more practical and efficient while achieving strong defense performance and adaptability.

A more detailed and extended review of related work is provided in App. A.

# 4 Methodology

In this section, we introduce the Bayesian Data Scheduler method. Specifically, we outline our framework in Sec. 4.1. Following that, we detail two implementations of BDS, including Bayesian Scalar Scheduler and Amortized Bayesian Neural Scheduler in Secs. 4.2 and 4.3.

## 4.1 Bayesian Data Scheduler Framework

We frame the harmful fine-tuning defense as a Bayesian inference framework, with its underlying graphical model depicted in Fig. 2. For clarity, the discussion here focuses on the left panel, with the right panel deferred to Sec. 4.3. Each data point $z_i^{\text{ft}}$ in the fine-tuning dataset $\mathcal{D}_{\text{ft}}$ is associated with another latent variable $w_i \in \mathbb{R}$, indicating its intrinsic safety attribute (*i.e.*, $z_i^{\text{ft}} \to w_i$). To mitigate the negative effect of potentially harmful data, the fine-tuning process $(w_i, z_i^{\text{ft}}) \to \boldsymbol{\theta}$ is constrained, where the contribution of each data point $z_i^{\text{ft}}$ to update the model $\boldsymbol{\theta}$

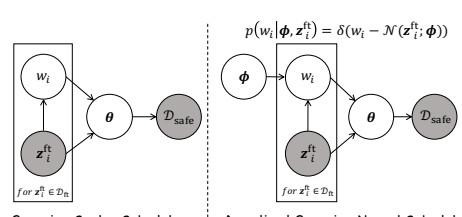

Figure 2: Graphical models for the Bayesian Scalar Scheduler (see Sec. 4.2) and Amortized Bayesian Neural Scheduler (see Sec. 4.3).

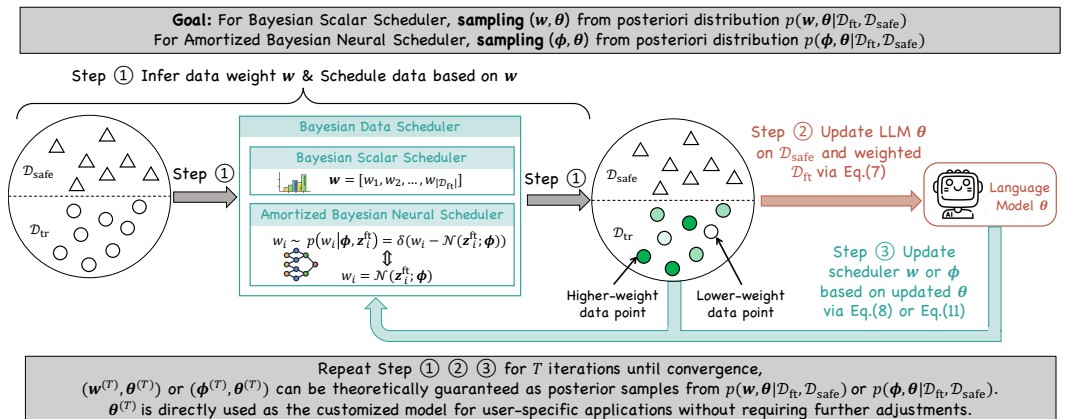

Figure 3: Pipeline of BDS. *Step 1:* BDS first infer the weight of each data point, indicating its safety attribute. *Step 2:* BDS updates the LLM $\boldsymbol{\theta}$ with weighted data via Eq. (7). *Step 3:* BDS update the scheduler $\boldsymbol{w}$ or $\phi$ via Eq. (8) or Eq. (11). Repeat steps 1-3 for $T$ iterations until convergence and $(\boldsymbol{w}^{(T)}, \boldsymbol{\theta}^{(T)})$ or $(\phi^{(T)}, \boldsymbol{\theta}^{(T)})$ can be theoretically guaranteed as posterior samples. $\boldsymbol{\theta}^{(T)}$ is directly used as the customized model for user-specific applications without requiring further adjustments. For clarity, the pseudocode for the BDS algorithm is provided in App. C.

is modulated by its safety attribute $w_i$. The edge $\boldsymbol{\theta} \to \mathcal{D}_{\text{safe}}$ represents the generation of the target parts of the alignment dataset $\mathcal{D}_{\text{safe}}$, modeling the likelihood of observing $\mathcal{D}_{\text{safe}}$ given $\boldsymbol{\theta}$. The data points are assumed to be i.i.d., implying that $p(\boldsymbol{w}) = \prod_i p(w_i)$ holds, where $w_i$ is the $i$-th component of vector $\boldsymbol{w}$. The conditional independence $\mathcal{D}_{\text{safe}} \perp (\boldsymbol{w}, \mathcal{D}_{\text{ft}}) \mid \boldsymbol{\theta}$ naturally holds from the structure of the graphical model.

**Goal.** Our goal is to infer the posterior distributions of unobserved variables (unshaded nodes): *i.e.*, the data weights $\boldsymbol{w}$ and the model parameters $\boldsymbol{\theta}$, conditioned on observed variables (shaded nodes), *i.e.*, the safety alignment dataset $\mathcal{D}_{\text{safe}}$ and the fine-tuning dataset $\mathcal{D}_{\text{ft}}$. Formally, this corresponds to estimating the joint posterior: $p(\boldsymbol{w}, \boldsymbol{\theta} \mid \mathcal{D}_{\text{ft}}, \mathcal{D}_{\text{safe}})$.

## 4.2 Bayesian Scalar Scheduler

**Posterior decomposition.** To address the computational intractability of directly inferring the posterior distribution $p(\boldsymbol{w}, \boldsymbol{\theta} \mid \mathcal{D}_{\text{ft}}, \mathcal{D}_{\text{safe}})$, we decompose it as follows:

$$p(\boldsymbol{\theta}, \boldsymbol{w} \mid \mathcal{D}_{\text{ft}}, \mathcal{D}_{\text{safe}}) \propto p(\mathcal{D}_{\text{safe}} \mid \boldsymbol{\theta}) \cdot p(\mathcal{D}_{\text{ft}} \mid \boldsymbol{\theta}, \boldsymbol{w}) \cdot p(\mathcal{D}_{\text{ft}} \mid \boldsymbol{w})^{-1} \cdot p(\boldsymbol{\theta}, \boldsymbol{w}). \tag{1}$$

Detailed derivations are provided in App. H.1. The term $p(\mathcal{D}_{\text{safe}} \mid \boldsymbol{\theta})$ in Eq. (1) represents the likelihood of observing the alignment dataset $\mathcal{D}_{\text{safe}}$ given the model parameters $\boldsymbol{\theta}$, quantifying how well the model aligns with the alignment dataset. Based on the conventional trick [82], this likelihood can be formulated in terms of the loss function as follows:

$$p(\mathcal{D}_{\text{safe}} \mid \boldsymbol{\theta}) \propto \prod_{i=1}^{|\mathcal{D}_{\text{safe}}|} \exp\left(-\ell\left(\boldsymbol{z}_i^{\text{safe}}; \boldsymbol{\theta}\right)\right), \tag{2}$$

where $|\mathcal{D}_{\text{safe}}|$ denotes the size of the dataset and $\ell(\cdot)$ denotes the loss function. Next, the second term $p(\mathcal{D}_{\text{ft}} \mid \boldsymbol{\theta}, \boldsymbol{w})$ in Eq. (1) represents the likelihood of the fine-tuning dataset $\mathcal{D}_{\text{ft}}$ given the model parameters $\boldsymbol{\theta}$ and the safety attributes $\boldsymbol{w}$, where the contribution of each data point $\boldsymbol{z}_i^{\text{ft}}$ is modulated by its safety weight $w_i$:

$$p(\mathcal{D}_{\text{ft}} \mid \boldsymbol{\theta}, \boldsymbol{w}) \propto \prod_{i=1}^{|\mathcal{D}_{\text{ft}}|} \exp\left(-\sigma(w_i) \cdot \ell\left(\boldsymbol{z}_i^{\text{ft}}; \boldsymbol{\theta}\right)\right), \tag{3}$$

where $\sigma(\cdot)$ refers to the weight transformation function, which is discussed in detail in Sec. 4.4.

The term $p(\mathcal{D}_{\text{ft}} \mid \boldsymbol{w})^{-1}$ in Eq. (1) represents the likelihood of the fine-tuning dataset $\mathcal{D}_{\text{ft}}$ given the safety attributes $\boldsymbol{w}$. Since it is infeasible to directly model the relationship $\mathcal{D}_{\text{ft}} \to \boldsymbol{w}$, we employ

marginalization over $\boldsymbol{\theta}$ and approximate it using the maximum a posteriori (MAP) estimate (Detailed derivations are provided in App. H.2.):

$$p\left(\mathcal{D}_{\mathrm{ft}} \mid \boldsymbol{w}\right)^{-1} \propto \prod_{i=1}^{|\mathcal{D}_{\mathrm{ft}}|} \exp\left(\sigma(w_i) \cdot \ell\left(\boldsymbol{z}_i^{\mathrm{ft}}; \hat{\boldsymbol{\theta}}\right)\right), \quad \text{s.t.,} \quad \hat{\boldsymbol{\theta}} = \arg\max_{\boldsymbol{\theta}} \mathbb{E}_{p(\mathcal{D}_{\mathrm{ft}}|\boldsymbol{w})}\left[\left(p\left(\boldsymbol{\theta} \mid \mathcal{D}_{\mathrm{ft}}, \boldsymbol{w}\right)\right].\quad (4)$$

**Efficient Posterior inference via Stochastic Gradient Langevin Dynamic (SGLD) sampling.** Due to the intractability of obtaining closed-form solutions for Eq. (1), we employ an efficient posterior inference technique based on stochastic-gradient Markov Chain Monte Carlo (SG-MCMC) [45]. Specifically, we utilize Stochastic Gradient Langevin Dynamics (SGLD) [69] to sample from the posterior $p(\boldsymbol{w}, \boldsymbol{\theta} \mid \mathcal{D}_{\mathrm{ft}}, \mathcal{D}_{\mathrm{safe}})$, inspired by [49, 17, 73, 10]. By iteratively updating the system under the Langevin dynamics framework, we can obtain theoretically guaranteed posterior samples once the process converges (convergence analysis is provided in App. G.2).

$$[\boldsymbol{\theta}, \boldsymbol{w}] \leftarrow [\boldsymbol{\theta}, \boldsymbol{w}] + \frac{\eta}{2}\nabla_{\boldsymbol{\theta}, \boldsymbol{w}} \log p\left(\boldsymbol{\theta}, \boldsymbol{w} \mid \mathcal{D}_{\mathrm{ft}}, \mathcal{D}_{\mathrm{safe}}\right) + \epsilon\sqrt{\eta}, \quad (5)$$

where $\eta$ represents the step size, and $\epsilon \sim \mathcal{G}(0, I)$ denotes Gaussian noise introduced to inject randomness. Based on the posterior decomposition in Eq. (1), the gradient of the log-posterior can be expressed as a summation:

$$\nabla_{\boldsymbol{\theta}, \boldsymbol{w}} \log p\left(\boldsymbol{\theta}, \boldsymbol{w} \mid \mathcal{D}_{\mathrm{ft}}, \mathcal{D}_{\mathrm{safe}}\right) = \nabla_{\boldsymbol{\theta}, \_} \log p(\mathcal{D}_{\mathrm{safe}} \mid \boldsymbol{\theta}) + \nabla_{\boldsymbol{\theta}, \boldsymbol{w}} \log p(\mathcal{D}_{\mathrm{ft}} \mid \boldsymbol{\theta}, \boldsymbol{w})$$
$$+\nabla_{\_, \boldsymbol{w}} \log p(\mathcal{D}_{\mathrm{ft}} \mid \boldsymbol{w})^{-1} + \nabla_{\boldsymbol{\theta}, \boldsymbol{w}} \log p(\boldsymbol{\theta}, \boldsymbol{w}), \quad (6)$$

where placeholder _ denotes independence between the gradient term and the corresponding variable. To improve the efficiency of gradient computation, SGLD substitutes the full-batch likelihood in with a minibatch approximation. Using minibatch $\mathcal{B}$, we reformulate Eq. (6) using the decompositions in Eqs. (2) to (4), yielding the following equations.

$$\boldsymbol{\theta} \leftarrow \boldsymbol{\theta} + \frac{\eta}{2}\nabla_{\boldsymbol{\theta}}\left(\log p(\boldsymbol{\theta} \mid \boldsymbol{w}) - \frac{|\mathcal{D}_{\mathrm{safe}}|}{|\mathcal{B}_{\mathrm{safe}}|}\sum_{\boldsymbol{z}_i^{\mathrm{safe}} \in \mathcal{B}_{\mathrm{safe}}} \ell(\boldsymbol{z}_i^{\mathrm{safe}}; \boldsymbol{\theta}) - \frac{|\mathcal{D}_{\mathrm{ft}}|}{|\mathcal{B}_{\mathrm{ft}}|}\sum_{\boldsymbol{z}_i^{\mathrm{ft}} \in \mathcal{B}_{\mathrm{ft}}}\left[\sigma(w_i) \cdot \ell(\boldsymbol{z}_i^{\mathrm{ft}}; \boldsymbol{\theta})\right]\right) + \epsilon\sqrt{\eta},$$
$$(7)$$

$$\boldsymbol{w} \leftarrow \boldsymbol{w} + \frac{\eta}{2}\nabla_{\boldsymbol{w}}\left(\log p(\boldsymbol{w}) - \frac{|\mathcal{D}_{\mathrm{ft}}|}{|\mathcal{B}_{\mathrm{ft}}|}\sum_{\boldsymbol{z}_i^{\mathrm{ft}} \in \mathcal{B}_{\mathrm{ft}}}\left[\sigma(w_i) \cdot (\ell(\boldsymbol{z}_i^{\mathrm{ft}}; \boldsymbol{\theta}) - \ell(\boldsymbol{z}_i^{\mathrm{ft}}; \hat{\boldsymbol{\theta}}))\right]\right) + \epsilon\sqrt{\eta}$$
$$\approx \boldsymbol{w} + \frac{\eta}{2}\nabla_{\boldsymbol{w}}\left(\log p(\boldsymbol{w}) - \frac{|\mathcal{D}_{\mathrm{ft}}|}{|\mathcal{B}_{\mathrm{ft}}|}\sum_{\boldsymbol{z}_i^{\mathrm{ft}} \in \mathcal{B}_{\mathrm{ft}}}\left[\sigma(w_i) \cdot \ell(\boldsymbol{z}_i^{\mathrm{ft}}; \boldsymbol{\theta})\right]\right) + \epsilon\sqrt{\eta},$$
$$(8)$$

where we approximate $\ell(\boldsymbol{z}_i^{\mathrm{ft}}; \hat{\boldsymbol{\theta}}) \approx 0$. This is justified as $\hat{\boldsymbol{\theta}} = \arg\max_{\boldsymbol{\theta}} \mathbb{E}_{p(\mathcal{D}_{\mathrm{ft}}|\boldsymbol{w})}\left[\left(p\left(\boldsymbol{\theta} \mid \mathcal{D}_{\mathrm{ft}}, \boldsymbol{w}\right)\right]$ (see Eq. (4)), which represents the MAP estimate of the model $\boldsymbol{\theta}$ given the dataset $\mathcal{D}_{\mathrm{ft}}$. Intuitively, $\hat{\boldsymbol{\theta}}$ is optimized to perform well on $\mathcal{D}_{\mathrm{ft}}$, resulting in a near-zero loss.

**Intuition behind weight update in Eq. (8) (see Fig. 4).** By omitting the prior term, noise, and constants, the update of the $i$-th component of $\boldsymbol{w}$ in the non-approximated Eq. (8) simplifies to:
$$w_i \leftarrow w_i - \left(\ell(\boldsymbol{z}_i^{\mathrm{ft}}; \boldsymbol{\theta}) - \ell(\boldsymbol{z}_i^{\mathrm{ft}}; \hat{\boldsymbol{\theta}})\right) \cdot \nabla_{w_i}\sigma(w_i).$$
In this form, the loss gap scales the gradient term $\nabla_{w_i}\sigma(w_i)$, where a larger loss gap leads to a significant reduction in $w_i$, ultimately resulting in a smaller weight. Within the loss gap, $\boldsymbol{\theta}$ is derived via Eq. (7), which optimizes over both $\mathcal{D}_{\mathrm{safe}}$ and the weighted $\mathcal{D}_{\mathrm{ft}}$, thus referred to as the safety-aware model. In contrast, $\hat{\boldsymbol{\theta}}$ is obtained through Eq. (4), fitting only the weighted $\mathcal{D}_{\mathrm{ft}}$, thus re-

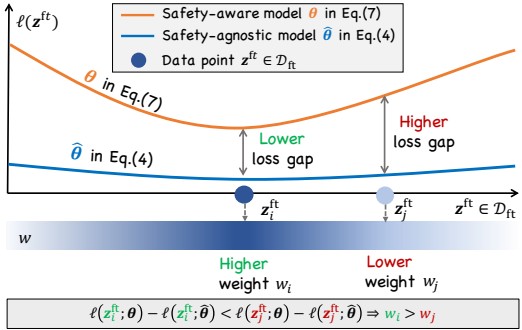

Figure 4: Intuition behind weight update in Eq. (8).

ferred to as the safety-agnostic model. If a data point $\boldsymbol{z}_i^{\mathrm{ft}}$ produces a large positive loss gap, it indicates poorer alignment with the safety-aware model $\boldsymbol{\theta}$ compared to the safety-agnostic model $\hat{\boldsymbol{\theta}}$. This suggests that $\boldsymbol{z}_i^{\mathrm{ft}}$ is likely misaligned with the safe dataset $\mathcal{D}_{\mathrm{safe}}$, ultimately resulting in a larger weight reduction, and thus a smaller assigned weight.

## 4.3 Amortized Bayesian Neural Scheduler

The Bayesian Scalar Scheduler introduced in Sec. 4.2 provides a strong foundation but leaves room for further enhancement in two key aspects: (i) Scalability: Inferring $w_i \sim p(w_i \mid z_i^{\text{ft}}, \mathcal{D}_{\text{safe}})$ for each data point scales with dataset size. (ii) Transferability: Without explicitly modeling $z_i^{\text{ft}} \to w_i$, posterior inference must restart from scratch when new data is added.

To address the scalability and transferability issues, we introduce the Amortized Bayesian Neural Scheduler based on amortized Bayesian learning [51]. The core idea is to amortize the effort for inferring datapoint-wise posterior $p(w_i \mid z_i^{\text{ft}}, \mathcal{D}_{\text{safe}})$ into inferring just one posterior of a neural network $p(\phi \mid \mathcal{D}_{\text{ft}}, \mathcal{D}_{\text{safe}})$. As shown in the right panel of Fig. 2, the neural network $\phi \sim p(\phi \mid \mathcal{D}_{\text{ft}}, \mathcal{D}_{\text{safe}})$ is shared across all data points, enabling each $p(w_i \mid z_i^{\text{ft}}, \mathcal{D}_{\text{safe}})$ to be inferred efficiently via a forward pass through the network conditioned on $z_i^{\text{ft}}$. For simplicity, we define $p(w_i \mid z_i^{\text{ft}}, \mathcal{D}_{\text{safe}})$ as a Dirac delta distribution $\delta(\cdot)$, whose single value is determined by the output of a neural network $\mathcal{N}(z_i^{\text{ft}} \mid \phi)$ with $z_i^{\text{ft}}$ as input. After introducing $\phi$, the datapoint-wise posterior is expressed as:

$$p(w_i \mid \phi, z_i^{\text{ft}}) := \delta\left(w_i - \mathcal{N}(z_i^{\text{ft}} \mid \phi)\right). \tag{9}$$

In this way, only a single posterior over the neural network parameters (*i.e.*, $p(\phi \mid \mathcal{D}_{\text{ft}}, \mathcal{D}_{\text{safe}})$) needs to be inferred, and the posterior of weights for new data can be inferred seamlessly through forward passes without retraining.

After introducing $\phi$, the updated probabilistic graphical model is illustrated in the right panel of Fig. 2. Consequently, our inference objective shifts from $p(w, \theta \mid \mathcal{D}_{\text{ft}}, \mathcal{D}_{\text{safe}})$ to $p(\phi, \theta \mid \mathcal{D}_{\text{ft}}, \mathcal{D}_{\text{safe}})$, which can be decomposed as follows (Detailed proof is provided in App. H.3):

$$p(\theta, \phi \mid \mathcal{D}_{\text{ft}}, \mathcal{D}_{\text{safe}}) \propto p(\mathcal{D}_{\text{safe}} \mid \theta) \cdot p(\mathcal{D}_{\text{ft}} \mid \theta, w) \cdot p(\mathcal{D}_{\text{ft}} \mid w)^{-1} \cdot p(\theta, \phi \mid \mathcal{D}_{\text{ft}}),$$

$$\text{s.t.,} \quad w = \mathcal{N}(\mathcal{D}_{\text{ft}} \mid \phi) \triangleq \left[\mathcal{N}(z_1^{\text{ft}} \mid \phi), \cdots, \mathcal{N}(z_{|\mathcal{D}_{\text{ft}}|}^{\text{ft}} \mid \phi)\right]. \tag{10}$$

Similar to Eq. (8), we derive the update rule for $\phi$:

$$\phi \leftarrow \phi + \frac{\eta}{2} \nabla_\phi \left( \log p(\phi) - \frac{|\mathcal{D}_{\text{ft}}|}{|\mathcal{B}_{\text{ft}}|} \sum_{z_i^{\text{ft}} \in \mathcal{B}_{\text{ft}}} \left[ \sigma(\mathcal{N}(z_i^{\text{ft}}; \phi)) \cdot \ell(z_i^{\text{ft}}; \theta) \right] \right) + \epsilon\sqrt{\eta}. \tag{11}$$

## 4.4 Data Weight Transformation

Recall the scheduler updates in Eqs. (8) and (11). We identify a key factor that significantly impacts the performance of BDS: the weight transformation function $\sigma(w_i)$. As shown in Fig. 5, different transformation functions result in varying sampling trajectories of $w$ for benign and harmful data, ultimately leading to distinct defensive capabilities (see Tab. 14). For clarity, we first introduce the definition of the SGLD Sampling Trajectory:

**Definition 4.1.** (*SGLD Sampling Trajectory*) The *SGLD Sampling Trajectory* refers to the sequence of states generated during the iterative sampling process of SGLD. This sequence forms a Markov chain, where each state transition depends solely on the previous state. For the distribution $p(w, \theta \mid \mathcal{D}_{\text{ft}}, \mathcal{D}_{\text{safe}})$, the trajectories of the variables $w$ and $\theta$ can be defined as:

$$\text{Tr}_w = w^{(0)} \to \cdots \to w^{(T)}, \ w^{(t+1)} = w^{(t)} + \frac{\eta}{2}\nabla_w^{(t)} + \epsilon\sqrt{\eta},$$

$$\text{Tr}_\theta = \theta^{(0)} \to \cdots \to \theta^{(T)}, \ \theta^{(t+1)} = \theta^{(t)} + \frac{\eta}{2}\nabla_\theta^{(t)} + \epsilon\sqrt{\eta}. \tag{12}$$

Here, we use $\nabla_w$ and $\nabla_\theta$ to denote the gradient terms in Eqs. (7) and (8). We next analyze the effect of different transformation functions on the term $\nabla_w$ in Eq. (12).

We consider three types of weight transformations: *identity*, *sigmoid*, and *softmax*, which are formally defined as:

$$\sigma(w_i) = w_i \quad (\textit{identity}), \quad \bar{\sigma}(w_i) = \frac{1}{1 + \exp(-w_i)} \quad (\textit{sigmoid}),$$

$$\bar{\bar{\sigma}}(w_i) = \frac{\exp(w_i)}{\sum_{j=1}^{|\mathcal{D}_{\text{ft}}|} \exp(w_j)} \quad (\textit{softmax}). \tag{13}$$

By differentiating $w$ in Eq. (8) (excluding prior and constant coefficients), we derive that under the identity transformation, $w$ updates in a monotonically decreasing manner:

$$\nabla_{w_i} = -\ell\left(z_i^{\text{ft}}; \theta\right) < 0. \tag{14}$$

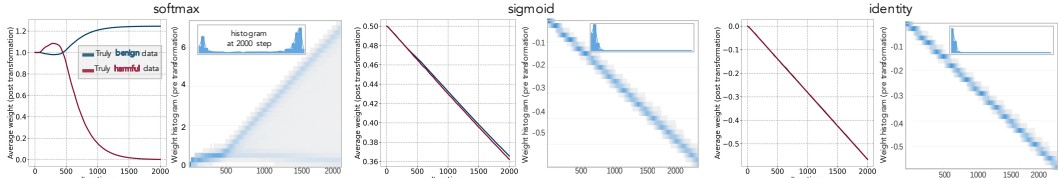

Figure 5: Effect of weight transformation on SGLD sampling trajectories of $w$ for benign and harmful data, respectively. For clarity, weights post-softmax are scaled by $|\mathcal{D}_{\text{safe}}|$.

Similarly, under sigmoid, the weight $w$ also exhibits a monotonically decreasing update behavior:

$$\nabla_{w_i} = -\bar{\sigma}(x) \cdot (1 - \bar{\sigma}(x)) \cdot \ell\left(\boldsymbol{z}_i^{\text{ft}}; \boldsymbol{\theta}\right) < 0. \tag{15}$$

Since the gradient of $w$ is proportional to the negative loss, the weight suboptimality under both identity and sigmoid transformation primarily arises from the uniform treatment of benign and harmful data, resulting in monotonically decreasing weights for both. As demonstrated in Fig. 5, this causes the scheduler to assign reduced weights to both benign and harmful data to preserve the model's inherent safety alignment, which is equivalent to discarding any data and consequently leads to poor fine-tuning performance.

In contrast, under softmax, the weight $w$ updates in an adaptively bidirectional manner:

$$\nabla_{w_i} = -\bar{\bar{\sigma}}(w_i) \cdot \left(\ell(\boldsymbol{z}_i^{\text{ft}}; \boldsymbol{\theta}) - \sum_{k=1}^{|\mathcal{D}_{\text{ft}}|} \bar{\bar{\sigma}}(w_k) \cdot \ell(\boldsymbol{z}_k^{\text{ft}}; \boldsymbol{\theta})\right) \begin{cases} > 0, & \text{if } \ell(\boldsymbol{z}_i^{\text{ft}}; \boldsymbol{\theta}) < \sum_{k=1}^{|\mathcal{D}_{\text{ft}}|} \bar{\bar{\sigma}}(w_k) \cdot \ell(\boldsymbol{z}_k^{\text{ft}}; \boldsymbol{\theta}) \\ < 0, & \text{if } \ell(\boldsymbol{z}_i^{\text{ft}}; \boldsymbol{\theta}) > \sum_{k=1}^{|\mathcal{D}_{\text{ft}}|} \bar{\bar{\sigma}}(w_k) \cdot \ell(\boldsymbol{z}_k^{\text{ft}}; \boldsymbol{\theta}) \end{cases}. \tag{16}$$

Detailed derivations of Eq. (16) is provided in App. H.4. Unlike identity and sigmoid transformation monotonically decreasing $w$ based solely on the absolute magnitude of the loss, the softmax updates weights based on the relative value of the loss. This ensures that the model does not assign extremely low weights to all data points. Specifically, the softmax transformation uses the weighted average loss as a reference: if the loss of a data point is below the reference, its weight increases; otherwise, its weight decreases. Moreover, since $\mathcal{D}_{\text{ft}}^{\text{benign}}$ aligns more closely with $\mathcal{D}_{\text{safe}}$ than $\mathcal{D}_{\text{ft}}^{\text{harmful}}$, the benign data exhibits relatively lower loss, leading to an increase in their weights (see Fig. 5).

**Time-weighted accumulation of posterior bias.** We further explore weight suboptimality under identity transformation through the theoretical perspective of posterior bias, a newly proposed concept with detailed formulation and analysis in Apps. I.1 to I.5. Here, we only present the core theorem below (Detailed proof is provided in App. I.6):

**Theorem 4.2** (Time-Weighted Accumulation of Posterior Bias). *Let* $(\text{Tr}_{\boldsymbol{w}}, \text{Tr}_{\boldsymbol{\theta}})$ *and* $(\text{Tr}_{\boldsymbol{w}^*}, \text{Tr}_{\boldsymbol{\theta}^*})$ *denote the SGLD sampling trajectories under identity transformation drawn from the target distributions* $p(\boldsymbol{w}, \boldsymbol{\theta} \mid \mathcal{D}_{\text{ft}}, \mathcal{D}_{\text{safe}})$ *and* $p(\boldsymbol{w}^*, \boldsymbol{\theta}^* \mid \mathcal{D}_{\text{ft}}, \mathcal{D}_{\text{safe}}, \mathcal{D}_{\text{ft}}^{\text{val}})$, *respectively. Here,* $\mathcal{D}_{\text{ft}}^{\text{val}}$ *is a held-out clean tuning dataset, serving as a test or validation set to evaluate fine-tuning performance. Then, the following proportionality holds:*

$$\underbrace{\mathbb{E}_{\text{Tr}_{\boldsymbol{w}}, \text{Tr}_{\boldsymbol{w}^*}} \left\| \boldsymbol{w}^{(T)} - \boldsymbol{w}^{*(T)} \right\|}_{\mathcal{PB}^{(T)}} \propto \underbrace{\mathbb{E}_{\text{Tr}_{\bar{\boldsymbol{w}}}, \text{Tr}_{\bar{\boldsymbol{w}}^*}} \sum_{t=0}^{T-1}(T-t) \left\| \boldsymbol{w}^{(t)} - \boldsymbol{w}^{*(t)} \right\|}_{\sum_{t=1}^{T-1}(T-t)\mathcal{PB}^{(t)}}. \tag{17}$$

*Here,* $\mathcal{PB}^{(T)}$ *quantifies the posterior bias at iteration* $T$, *and summation* $\sum_{t=1}^{T-1}(T-t)\mathcal{PB}^{(t)}$ *characterizes the cumulative posterior bias over the entire sampling trajectory.*

**Remark.** Notably, the time-weighted factor $T - t$ highlights the greater influence of earlier iterations on the cumulative bias, suggesting that suboptimal sampling in the early stages of the SGLD trajectory can propagate and aggressively impact the overall posterior inference.

## 5 Experiment

**Datasets and models.** Our experimental setup primarily follows [29, 33] to ensure fair comparison. For the alignment dataset ($\mathcal{D}_{\text{safe}}$, consisting of harmful prompt-safe answer pairs), we use the dataset

Table 1: Comparison with SOTA baselines, ($|\mathcal{D}_{\text{safe}}| = 1000$ and $|\mathcal{D}_{\text{ft}}| = 1000$ for BDS).

| Method | $\mathcal{D}_{\text{safe}}$ | $\mathcal{D}_{\text{harmful}}$ | Harmful Score ↓ | | | | | | Finetune Accuracy ↑ | | | | | |
|---|---|---|---|---|---|---|---|---|---|---|---|---|---|---|
| | | | clean | p=0.05 | p=0.1 | p=0.15 | p=0.2 | Average | clean | p=0.05 | p=0.1 | p=0.15 | p=0.2 | Average |
| SFT | ✓ | ✗ | 1.30 | 21.90 | 33.70 | 49.30 | 61.70 | 33.58 | 81.54 | 91.74 | 93.12 | 92.66 | 92.89 | 90.39 |
| Lisa | ✓ | ✗ | 0.90 | 14.50 | 23.7 | 31.20 | 39.10 | 21.88 | 86.93 | 91.86 | 92.32 | 92.20 | 92.32 | 91.13 |
| Repnoise | ✓ | ✓ | 1.20 | 20.70 | 32.10 | 45.60 | 55.50 | 31.02 | 90.25 | 92.89 | 93.00 | 92.89 | 92.89 | 92.38 |
| Vaccine | ✓ | ✓ | 1.30 | 12.10 | 28.3 | 44.10 | 55.20 | 28.20 | 90.83 | 93.58 | 93.69 | 93.23 | 93.23 | 92.91 |
| Booster | ✓ | ✓ | 1.90 | 4.80 | 8.30 | 14.20 | 25.50 | 10.94 | 92.89 | 92.32 | 93.23 | 93.35 | 93.35 | 93.03 |
| BDS | ✓ | ✗ | 1.10 | **1.60** | **1.20** | **1.50** | **1.30** | **1.34** | **93.81** | **94.04** | **93.69** | **93.92** | **93.69** | **93.83** |

Table 2: Robustness for large harmful ratios $p$.

| Method | Harmful Score ↓ | | | | | | | | Finetune Accuracy ↑ | | | | | | | |
|---|---|---|---|---|---|---|---|---|---|---|---|---|---|---|---|---|
| | p=0.3 | p=0.4 | p=0.5 | p=0.6 | p=0.7 | p=0.8 | p=0.9 | p=1.0 | p=0.3 | p=0.4 | p=0.5 | p=0.6 | p=0.7 | p=0.8 | p=0.9 | p=1.0 |
| Booster | 40.60 | 68.40 | 77.20 | 76.90 | 77.50 | 76.30 | 75.90 | 76.60 | 92.12 | 93.00 | 92.69 | 92.20 | 91.63 | 91.63 | 91.51 | — |
| BDS | **1.20** | **1.50** | **1.20** | **1.50** | **1.50** | **1.60** | **1.50** | **1.80** | **93.32** | **93.19** | **93.00** | **92.66** | **92.29** | **92.78** | **92.89** | — |

from [53], an enriched version of BeaverTails [36]. To simulate harmful fine-tuning attack, the fine-tuning dataset ($\mathcal{D}_{\text{ft}}$) is constructed by mixing a proportion $p$ of unsafe data ($\mathcal{D}_{\text{ft}}^{\text{harmful}}$) from BeaverTails with $1 - p$ benign fine-tuning data ($\mathcal{D}_{\text{ft}}^{\text{benign}}$), resulting in a total size of $|\mathcal{D}_{\text{ft}}|$. Fine-tuning tasks are considered on SST2 [56], AGNEWS [80], GSM8K [9], AlpacaEval [38], and GEM benchmark. Model architectures include Llama2-7B [60], Gemma2-9B [59] and Qwen2-7B [74]. Default settings are $p = 0.1$ and $|\mathcal{D}_{\text{ft}}| = 1000$ ($|\mathcal{D}_{\text{ft}}| = 700$ for AlpacaEval).

**Metrics.** We adopt two evaluation metrics for assessing model performance, following [29, 33]:

- Finetune Accuracy (FA): The accuracy on the testing dataset of the corresponding fine-tuning task. Details on evaluation procedure are provided in App. B.2.

- Harmful Score (HS): Using the moderation model from [36], we classify model outputs as harmful or non-harmful. The harmful score is defined as the proportion of unsafe outputs.

For harmful score calculation, we sample 1000 instructions from the testing set of BeaverTails [36]. Finetune accuracy is evaluated using 872, 1000, 1000, 1000 and 104 samples from the fine-tuning datasets SST2, AGNEWS, GSM8K, GEM, and AlpacaEval, respectively.

**Baselines.** We compare our method with five representative defense baselines, including the SOTA Booster [29], Vaccine [33], Repnoise [52], Lisa [31], and SFT (Supervised Fine-Tuning) [29]. Descriptions and implementation details of each baseline are provided in App. B.1.

**Implementation details.** For fine-tuning, we adopt LoRA [22] for efficient fine-tuning, following [29, 33, 21]. Fine-tuning is performed using the FusedAdam [50] with a learning rate of $1 \times 10^{-5}$ and a weight decay of 0.1, as recommended by [29]. The training involves 20 epochs with a batch size of 10. The neural scheduler is implemented using a lightweight 125M Fairseq-Dense model [2] with an added trainable linear head. The scheduler's learning rate is set to $5 \times 10^{-3}$ for scalar scheduler and $1 \times 10^{-6}$ for neural scheduler. Unless otherwise specified, we use the scalar scheduler as default. More implementation details are provided in Apps. B.3 and B.4.

## 5.1 Main Results

**Comparison with SOTA defense baselines.** Tab. 1 presents comparison of BDS with several SOTA baselines across harmful ratios ranging from 0 to 0.2. BDS consistently outperforms all baselines in both defensive and fine-tuning performance, achieving a significant 9.60% reduction in average Harmful Score and a 0.8% improvement in average Finetune Accuracy. Existing baselines struggle to consistently mitigate harmful data influence: Simulation-based baselines like Booster and Vaccine enhance robustness by attack simulation, whose effectiveness diminishes significantly as the harmful ratio increases; RepNoise attempts to unlearn harmful knowledge but it inevitably recovers under higher harmful ratios; Lisa mixes alignment data with fine-tuning data but fundamentally lacks a deweighting mechanism and thus requires alignment data to scale with the unknown harmful ratio.

Notably, when the harmful ratio is 0 (*i.e.*, benign attack [48]), BDS achieves the highest and well-preserved fine-tuning performance, whereas SFT suffers significant drops. This can be explained by the *helpfulness–harmlessness trade-off*: SFT endows the model with strong harmlessness after alignment but reduces its plasticity [11] to sufficiently adapt and learn helpfulness during fine-tuning, thus leading to lower fine-tuning accuracy. As the harmful ratio increases (e.g., $p : 0 \rightarrow 0.2$), fine-tuning accuracy of SFT even rises because the introduced harmful data act as counterexamples

Table 3: Robustness across fine-tuning datasets ($|\mathcal{D}_{\text{safe}}| = 1000$, $|\mathcal{D}_{\text{ft}}| = 1000$, $p = 0.1$).

| Method | SST2 | | AGNews | | GSM8K | | Alpaca | |
|---|---|---|---|---|---|---|---|---|
| | HS ↓ | FA ↑ | HS ↓ | FA ↑ | HS ↓ | FA ↑ | HS ↓ | FA ↑ |
| SFT | 33.70 | 93.12 | 30.70 | 85.90 | 14.80 | 15.20 | 40.70 | 45.67 |
| Lisa | 23.70 | 92.32 | 16.80 | 83.20 | 5.10 | 12.00 | 14.30 | 41.35 |
| Repnoise | 32.10 | 93.00 | 27.30 | 85.50 | 16.60 | 16.10 | 36.50 | 41.83 |
| Vaccine | 28.30 | 93.69 | 25.20 | 86.10 | 3.70 | 15.30 | 43.40 | 44.71 |
| Booster | 8.30 | 93.23 | 7.10 | 87.20 | 6.40 | 17.10 | 36.70 | 45.19 |
| BDS | **1.20** | **93.69** | **1.10** | **89.10** | **2.00** | **18.30** | **2.30** | **48.08** |

Table 4: Robustness across model architectures ($|\mathcal{D}_{\text{safe}}| = 1000$, $|\mathcal{D}_{\text{ft}}| = 1000$, $p = 0.1$).

| Method | Llama2 | | Gemma2 | | Qwen2 | | Avg. | |
|---|---|---|---|---|---|---|---|---|
| | HS ↓ | FA ↑ | HS ↓ | FA ↑ | HS ↓ | FA ↑ | HS ↓ | FA ↑ |
| SFT | 33.70 | 93.12 | 64.30 | 94.50 | 25.50 | 94.84 | 41.17 | 94.15 |
| Lisa | 23.70 | 92.32 | 30.80 | 94.04 | 9.50 | 93.92 | 21.33 | 93.43 |
| Repnoise | 32.10 | 93.00 | 63.60 | 94.50 | 33.90 | 94.61 | 43.20 | 94.04 |
| Vaccine | 28.30 | 93.69 | 45.00 | 93.69 | 16.80 | 92.55 | 30.03 | 93.31 |
| Booster | 8.30 | 93.23 | 11.20 | 93.69 | 1.60 | **95.64** | 7.03 | 94.19 |
| BDS | **1.20** | **93.69** | **1.30** | **94.50** | **0.90** | 94.72 | **1.13** | **94.30** |

Table 5: Robustness for different $|\mathcal{D}_{\text{ft}}|$.

| Method | Harmful Score ↓ | | | | Finetune Accuracy ↑ | | | |
|---|---|---|---|---|---|---|---|---|
| | 500 | 1000 | 1500 | 2000 | 500 | 1000 | 1500 | 2000 |
| Booster | 3.80 | 8.30 | 20.10 | 33.60 | 92.66 | 93.23 | **94.04** | 94.15 |
| BDS | **1.20** | **1.20** | **0.90** | **1.20** | **93.35** | **93.69** | 93.12 | **94.28** |

Table 6: Robustness for different $|\mathcal{D}_{\text{safe}}|$.

| Method | Harmful Score ↓ | | | | Finetune Accuracy ↑ | | | |
|---|---|---|---|---|---|---|---|---|
| | 100 | 500 | 1000 | 1500 | 100 | 500 | 1000 | 1500 |
| Booster | 62.20 | 34.60 | 8.30 | 8.10 | 93.23 | 93.42 | 93.23 | 92.86 |
| BDS | **1.70** | **1.20** | **1.20** | **1.30** | **93.58** | **93.92** | **93.69** | **93.69** |

Table 7: Effectiveness and transferability of neural scheduler. $A \to B$ denotes that the neural scheduler is trained on dataset $A$ and directly applied to assign data weights on dataset $B$. [HS/FA](new_ID/OOD) denote metrics under in-domain and out-of-domain transfer settings.

| Method | SST2 | | SST2 → SST2 (unseen) | | SST2 → AGNEWS | |
|---|---|---|---|---|---|---|
| | HS | FA | HS(new_ID) | FA(new_ID) | HS(new_OOD) | FA(new_OOD) |
| Scalar Scheduler | 1.20 | 93.69 | — | — | — | — |
| Neural Scheduler | 1.70 | 93.69 | 2.50 | 93.23 | 2.80 | 89.20 |

that prompt the model to "unlearn" part of its harmlessness, thereby restoring plasticity and improving its capacity to learn helpfulness. Our method achieves a better helpfulness–harmlessness trade-off by jointly optimizing the safety alignment and fine-tuning objectives, as shown in Eq. (7). This joint optimization resembles a multi-task learning paradigm, which more effectively balances the competing objectives of harmlessness and helpfulness than the sequential learning paradigm used in SFT. Further discussions on the helpfulness–harmlessness trade-off are provided in App. F.2.

**Robustness for high harmful ratios.** Prior evaluations [29, 33] are limited to low harmful ratios, as existing methods struggle under high harmful ratios. To highlight our robustness, we conduct experiments under high harmful ratios ranging from 0.3 to 1.0. As shown in Tab. 2, BDS significantly surpasses the SOTA Booster across all tested ratios in both harmful score and fine-tuning accuracy. At high harmful ratios from 0.5 to 1.0, BDS achieves a significant reduction in harmful score by over 70% and delivers a near 1% improvement in fine-tuning accuracy. These results highlight the exceptional robustness and reliability of BDS, even under highly adversarial conditions.

**Robustness for diverse fine-tuning tasks.** To assess generalization, we evaluate BDS on four fine-tuning datasets in Tab. 3. Results show that BDS consistently maintains exceptionally low harmful scores (around 1) across all tasks, while existing baselines exhibit severe instability and poor generalization. For instance, while Vaccine achieves a harmful score of 3.7 on GSM8K, it catastrophically underperforms on AlpacaEval with a harmful score of 43.30. These results highlight BDS's adaptability and robustness to complex tasks like GSM8K and AlpacaEval while ensuring harmlessness. More experiments on complex text generation tasks are provided in App. D.6.

**Robustness for different model architectures.** To further assess generalization, we evaluate BDS on two SOTA architectures, Gemma2-9B and Qwen2-7B. Tab. 4 demonstrates BDS consistently achieves low harmful scores, while existing baselines exhibit severe instability. While Booster attains a harmful score of 1.6 on Qwen2-7B, it collapses on Gemma2-9B, surging to 11.2. These highlight BDS's adaptability, ensuring consistent SOTA performance across diverse model architectures.

**Robustness for different sizes of $|\mathcal{D}_{\text{ft}}|$.** To validate robustness to $|\mathcal{D}_{\text{ft}}|$ size, we evaluate its impact while keeping $|\mathcal{D}_{\text{safe}}|$ and the harmful ratio $p$ fixed by default. As shown in Tab. 5, existing baseline suffers a sharp decline in defensive performance as $|\mathcal{D}_{\text{ft}}|$ increases. Booster's harmful score surges dramatically from 3.8 to 33.6, exposing its inability to handle larger datasets. In contrast, BDS demonstrates strong robustness, maintaining a consistently low harmful score of around 1 regardless of $|\mathcal{D}_{\text{ft}}|$ size. This highlights BDS's reliability in defending harmful fine-tuning, even as dataset sizes scale substantially. Robustness for larger dataset size is also demonstrated in App. D.5.

**Robustness for different sizes of $|\mathcal{D}_{\text{safe}}|$.** To validate robustness to $|\mathcal{D}_{\text{safe}}|$ size, we evaluate its impact while keeping $|\mathcal{D}_{\text{ft}}|$ and $p$ fixed by default. As shown in Tab. 6, existing baselines collapses as $|\mathcal{D}_{\text{safe}}|$ decreases, with defensive performance deteriorating sharply. In contrast, BDS exhibits remarkable

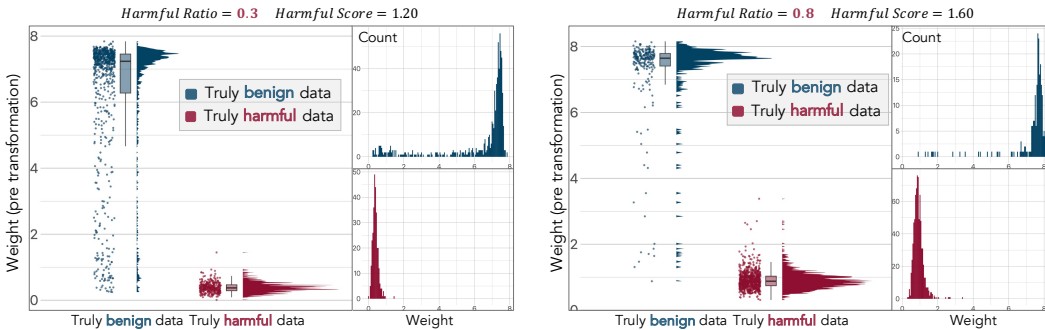

Figure 6: Data weights distributions across varying harmful ratios. More visualizations of weight distributions and scheduling dynamics are provided in Fig. 7 of App. D.1 and Fig. 8 of App. D.2.

resilience, consistently maintaining a harmful score below 2, even with only 100 alignment samples. These results underscore BDS's robustness in defense, even under limited alignment data.

**Effectiveness and transferability of neural scheduler.** To evaluate the effectiveness of the neural scheduler, we compare it with the scalar scheduler. The results in Tab. 7 show that the neural scheduler performs comparably to the scalar scheduler in both defensive and fine-tuning performance, achieving a harmful score below 2 on SST2. To further evaluate its transferability, we conduct experiments on unseen in-domain and out-of-domain datasets. The neural scheduler trained on the seen data is directly used to assign data weights to unseen data, which is subsequently employed for fine-tuning the LLM. As shown in Tab. 7, the neural scheduler generalizes effectively to both in-domain and out-of-domain unseen data, maintaining a harmful score below 3. These results highlight the strong effectiveness and transferability of the neural scheduler without the need for retraining, supporting its capacity to learn transferable data safety attributes [16].

**Robustness for advanced attack: OOD and ISA.** To evaluate the superior adaptiveness of our method under diverse attack dynamics, we conduct experiments against several challenging attack strategies: OOD attacks, and identity shifting attacks (ISA) [48]. Detailed results and discussions are provided in Apps. D.3 and D.4 due to limited space. Our approach significantly outperforms SOTA baselines, maintaining a low harmfulness ratio (around 1) across all attack strategies. This adaptiveness fundamentally stems from our loss-based data scheduling mechanism. Regardless of the attacker's strategy—whether OOD or ISA—harmful data tends to exhibit shared "unsafe behavior" (*i.e.*, incurring higher loss within the loss landscape of safety-aware models [46]), thus being assigned with lower weights. This also mirrors the shared "safe behavior" observed across benign datasets (see Tab. 7), and supports the existence of transferable data safety attributes discussed in [16]. This mechanism enables our method to effectively adapt to various attack dynamics without explicit attack simulation.

**Visualization of adaptiveness.** Weight distributions (Fig. 6 and Fig. 7 in App. D.1) and scheduling dynamics (Fig. 8 in App. D.2) demonstrates BDS adaptively and correctly assigns higher weights to truly benign data and lower weights to truly harmful data, across varying harmful ratios.

**Ablation studies** on each component configuration are provided in detail in Apps. E.1 to E.3 and E.5.

**Discussion.** We offer comprehensive discussions on the adaptiveness of BDS, helpfulness-harmlessness trade-offs, limitations, and social impact in Apps. F.1 to F.5, F.7 and F.8, along with complexity and convergence analyses in Apps. G.1 and G.2.

## 6 Conclusion

We propose BDS, a novel and principled framework that achieves adaptive defense against harmful fine-tuning without requiring attack simulation. We introduce two implementations, Bayesian Scalar Scheduler and Amortized Bayesian Neural Scheduler, with the latter enabling efficient transfer to new data without retraining. Leveraging the post hoc nature of Bayesian inference, BDS learns the posterior distribution of data weights conditioned on the specific dataset, thus achieving adaptive defense. Extensive experiments confirm the SOTA performance and superior adaptiveness of BDS, significantly outperforming baselines by over 70% across diverse attack and defense settings.

## Acknowledgments

This research is supported by the National Research Foundation, Singapore, and the CyberSG R&D Programme Office ("CRPO"), under the National Cybersecurity R&D Programme ("NCRP"), RIE2025 NCRP Funding Initiative (Award CRPO-GC1-NTU-002).

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

# Appendix of Adaptive Defense against Harmful Fine-Tuning for Large Language Models via Bayesian Data Scheduler

## Contents

## A    Related Work

Fine-tuning-as-a-service has become a widely adopted paradigm among mainstream LLM providers (*e.g.*, OpenAI[4] and Mistral[5]). Recent red teaming studies [48, 29, 30] have revealed a critical vulnerability: harmful fine-tuning. Harmful fine-tuning refers to that the presence of even a small fraction of harmful data in user- provided datasets can cause fine-tuned models to deviate from the safety alignment established during pre-training, *i.e.*, the model forgets to give refusal answer towards harmful prompts after fine-tuning on a few harmful samples.

**Mechanism study of harmful fine-tuning.** Existing research has made efforts to analyze the mechanisms underlying the high sensitivity of LLMs to harmful fine-tuning. (i) **Adversarial permutations:** One line of work reveals that harmful fine-tuning can induce adversarial perturbations to the model, to which LLMs are highly sensitive. For instance, Vicanne [33] demonstrates that tuning with harmful data in user-provided datasets can cause *embedding drift*. Similarly, Booster [29] identifies that harmful fine-tuning leads to *parameter perturbations*. Moreover, [46] introduces the concept of a *safety basin*, wherein LLMs can tolerate parameter perturbations within a local region while maintaining safety alignment; however, exceeding this basin results in a sharp degradation of alignment. (ii) **Structural vulnerabilities**: Another line of work delves deeper into the structural susceptibilities of specific model components [21, 47, 37]. For example, Safe LoRA [21] highlights that certain layers of a model play a more critical role in preserving safety, while others are less susceptible to perturbations. Similarly, [37] emphasizes that different layers perform distinct functions when exposed to various types of attacks. (iii) **Catastrophic forgetting**: Several studies explain alignment vulnerabilities through the lens of catastrophic forgetting [43] due to the sequential training paradigm [6, 13]. For instance, [63] highlights that the inconsistency between SFT and alignment objectives can lead to alignment knowledge being forgotten when SFT is performed sequentially after alignment. Likewise, [13] identifies that the sequential nature of SFT after alignment exacerbates this forgetting issue.

**Harmful fine-tuning attack.** On the attack side, [48] conduct experiments using OpenAI's API and demonstrate that even fine-tuning solely on benign data can compromise the base model's safety

---

[4]Fine-tuning API by OpenAI: https://platform.openai.com/docs/guides/fine-tuning
[5]Fine-tuning API by Mistral: https://docs.mistral.ai/guides/finetuning

alignment. This highlights the inherent risks of fine-tuning in the fine-tuning-as-a-service scenario. Further, [20] identify subsets of benign data that are more likely to degrade model safety post-fine-tuning. These data points are characterized by their proximity to harmful examples and distance from benign ones in both representation and gradient space. Additionally, [18] introduce the concept of *covert malicious fine-tuning*. In the first stage (learning the encoding), the model is trained to learn an encoding which it did not previously know. In the second stage, the model is fine-tuned with encoded harmful inputs and outputs. During testing, the model generates encoded harmful responses when triggered by encoded harmful queries. Recently, [32] construct harmful data designed to deliberately circumvent moderation guardrails.

**Harmful fine-tuning defense.** On the defense side, existing defense methods can be categorized into three categories, including alignment stage [33, 52, 58, 42, 29, 40, 41, 7, 75], fine-tuning stage [44, 3, 31, 61, 79, 55], and post-fine-tuning stage solutions [76, 12, 28, 70, 62]. Alignment stage defense aims at preemptively improving the model's robustness prior to deployment. For example, Vaccine [33] introduces simulated embedding perturbations to strengthen the model's embedding resistance, while Booster [29] incorporates parameter perturbations to achieve parameter resistance. RepNoise [52] unlearns harmful representations such that it is difficult to recover them during fine-tuning. Fine-tuning stage defense aims to reduce the influence of harmful data during the fine-tuning process. SafeInstr [4] proposes to mix safety alignment data during the fine-tuning process to constantly reinforce the model's alignment knowledge. Similarly, VLGuard [83] also employs the data-mixing strategy but focuses on verifying its effectiveness with Vision-LLMs. Lisa [31] also mixes alignment and fine-tuning data but introduces Bi-State Optimization to separate the optimization processes for alignment and fine-tuning data, thus reducing optimization overhead. As we can see, SafeInstr [4], VLGuard [83], and Lisa [31] similarly adopt the strategy of mixing alignment data with fine-tuning data. However, it has fundamental limitations: (i) These methods do not explicitly isolate harmful data but only attempt to counteract its effects indirectly, thus leading to suboptimal defense performance. (ii) They require alignment data to scale with harmful data, incurring high computational costs and making them highly sensitive to the unknown amount of harmful data in post hoc attacks. Seal [55] introduces a data selection strategy grounded in penalty-based bi-level optimization principles. The method assigns hard labels to data samples and relies on a manually adjusted threshold, which can be suboptimal since the defender generally lacks prior knowledge of the proportion of harmful data and the method could be less robust when dealing with ambiguous or uncertain samples (see discussions in App. F.4). Post-fine-tuning stage defense [76, 12, 28, 70, 62] aims to restore safety alignment after fine-tuning without sacrificing fine-tuning performance.

**Data curation for LLMs.** Data curation methods aim to optimize data utilization strategies. For instance, coreset selection for LLMs [81, 1, 34, 73, 41] focuses on selecting the most critical subset of data. Existing approaches typically rely on metrics computed from the raw dataset and a reference dataset, including gradient matching [71], representation similarity [19], influence functions [39, 78, 77], and uncertainty estimation [34]. However, these methods assume distributional similarity between the raw and reference datasets and often leads to high computational costs (*e.g.*, influence functions). In our setting, where the raw dataset (*i.e.*, user-provided fine-tuning dataset) and reference dataset (*i.e.*, alignment dataset) exhibit insufficient distributional similarity, such methods lead to suboptimal defense performance. In contrast, our approach leverages the loss difference between benign and harmful data on a model trained with the alignment dataset. This design is independent of distributional similarity and computationally efficient, thus offering a unique advantage in the setting of harmful fine-tuning defense.

# B  More Details

## B.1  Details on Baselines

We summarize the high-level ideas and implementation details of the baseline methods used in our experiments. We basically follow the configuration used in their original paper.

**SFT:** Supervised fine-tuning (SFT) is initially applied using the alignment dataset to fine-tune the base model, followed by SFT on the user-provided fine-tuning dataset, which contains partially harmful data.

**Vaccine [33]:** Vaccine introduces simulated perturbations to embeddings using a harmful dataset to enhance the model's robustness against influences of harmful data. Afterward, SFT is performed on the user-provided fine-tuning dataset, which includes partially harmful data. The hyperparameter $\rho$ is selected through grid search over $\{0.1, 1, 2, 5, 10\}$, with $\rho = 5$ used in the final experiments.

**RepNoise [52]:** RepNoise unlearns information about harmful representations such that it is difficult to recover them during fine-tuning. The hyperparameters are set as $\alpha = 1$ and $\beta = 0.001$.

**Lisa [31]:** Lisa alternatively tune the base model between alignment and fine-tuning datasets to preserve alignment knowledge. The regularizer intensity $\rho$ is selected via grid search over $\{0.001, 0.01, 0.1, 1\}$, with $\rho = 0.01$.

**Booster [29]:** Booster introduces simulated perturbations to model parameters using a harmful dataset, enhancing the model's robustness against potential harmful data by improving its resistance to parameter perturbations. Subsequently, Booster applies SFT to fine-tune the base model on the user-provided fine-tuning dataset.

### B.2 Details on Fine-Tuning Task Evaluation

For a fair comparison [29], we employ a unified system prompt for training and testing across all tasks, structured as follows:

> **Prompt:** Below is an instruction that describes a task, paired with an input that provides further context. Write a response that appropriately completes the request. Instruction:{`instruction`} Input:{`input`} Response:
> **Output:** {`output`}

We define {`instruction`} and {`input`} tailored to each dataset. For SST2, the {`instruction`} specifies sentiment analysis objective, with the {`input`} being a sentence and the {`output`} the corresponding sentiment label. In GSM8K, the {`instruction`} is a mathematical question, and the {`output`} is the correct answer. For AGNEWS, the {`instruction`} specifies the classification objective, with the {`input`} being a sentence and the {`output`} representing the corresponding category. For AlpacaEval, GPT4's high-quality instruction-answer pairs are used as the demonstration data, and testing involves evaluating the helpfulness of model responses to unseen prompts using ChatGPT's API.

### B.3 Details on Scheduler Training

Due to limited prior knowledge about the dependencies, we assume factorized priors: $p(\boldsymbol{\theta}, \boldsymbol{w}) \approx p(\boldsymbol{\theta}) \cdot p(\boldsymbol{w})$ in Eq. (1) and $p(\boldsymbol{\theta}, \boldsymbol{\phi} \mid \mathcal{D}_{\text{ft}}) \approx p(\boldsymbol{\theta}) \cdot p(\boldsymbol{\phi})$ in Eq. (10). Although $p(\boldsymbol{w})$ is theoretically defined as a prior distribution, in practice we initialize $\boldsymbol{w}$ to 0.1 and allow it to be freely optimized during training (i.e., using a noninformative prior) to avoid potentially misleading manually chosen priors. Future work could explore how to design more robust priors to better regularize the data weights. $p(\boldsymbol{\theta})$ and $p(\boldsymbol{\phi})$ follow zero-mean Gaussian distributions corresponding to weight decay regularization. The neural scheduler is implemented using a lightweight 125M Fairseq-Dense model [2] with an added trainable head. We adopt a identity transformation on the mean pooling of the instance's representations along the sequence length. The size of hidden state is 768. We optimizer the scalar and neural scheduler with a learning rate of $5 \times 10^{-3}$ and $1 \times 10^{-6}$, respectively, using a batch size of 10.

### B.4 Details on Fine-Tuning

For fine-tuning, we adopt LoRA [22] for efficient fine-tuning, following [29, 33, 21]. The adaptor rank is set to 32 with alpha set to 4. Fine-tuning is performed using the FusedAdam [50] with a learning rate of $1 \times 10^{-5}$ and a weight decay of 0.1, as recommended by [29]. The training involves 20 epochs for SST2, AGNEWS and GSM8K, and 100 epoches for AlpacaEval, following [29]. The batch size is set as 10. The model's backbone utilizes BF16 (bfloat16) precision for computational efficiency. For the extreme case where the harmful ratio is 1 (i.e., the fine-tuning data contain only harmful samples), we report in Tab. 2 the result obtained with a larger safe dataset ($|\mathcal{D}_{\text{safe}}| = 5000$) to achieve more stable defense performance, while results for other ratios use $|\mathcal{D}_{\text{safe}}| = 1000$.

## C   Algorithms

To provide a clearer understanding of our algorithm, we present detailed pseudocode in Alg. 1.

---

**Algorithm 1:** Bayesian Data Scheduler.

---

**Input:** Base LLM $\boldsymbol{\theta}^{(0)}$; User-provided fine-tuning dataset $\mathcal{D}_{\text{ft}}$; Alignment datset $\mathcal{D}_{\text{safe}}$; Bayesian Scalar Scheduler $\boldsymbol{w}$ or Amortized Bayesian Neural Scheduler $\mathcal{N}(\cdot; \boldsymbol{\phi})$; Step size $\eta$; Weight transformation function $\sigma(\cdot)$; Gaussian noise $\epsilon$; Max iterations $T$.

---

Initialize $\boldsymbol{w}$ or $\boldsymbol{\phi}$
**for** $t \leftarrow 0$ **to** $T$ **do**
   // Construct a batch of data
   Sample $\mathcal{B}_{\text{safe}}$ from the alignment dataset $\mathcal{D}_{\text{safe}}$ and $\mathcal{B}_{\text{ft}}$ from the fine-tuning dataset $\mathcal{D}_{\text{ft}}$
   // Update LLM via Eq. (7)

$$\boldsymbol{\theta}^{(t+1)} \leftarrow \boldsymbol{\theta}^{(t)} + \frac{\eta}{2} \nabla_{\boldsymbol{\theta}} \bigg( \log p(\boldsymbol{\theta}^{(t)} \mid$$

$$\boldsymbol{w}) - \frac{|\mathcal{D}_{\text{safe}}|}{|\mathcal{B}_{\text{safe}}|} \sum_{\boldsymbol{z}_i^{\text{safe}} \in \mathcal{B}_{\text{safe}}} \ell(\boldsymbol{z}_i^{\text{safe}}; \boldsymbol{\theta}^{(t)}) - \frac{|\mathcal{D}_{\text{ft}}|}{|\mathcal{B}_{\text{ft}}|} \sum_{\boldsymbol{z}_i^{\text{ft}} \in \mathcal{B}_{\text{ft}}} \Big[ \sigma(w_i) \cdot \ell(\boldsymbol{z}_i^{\text{ft}}; \boldsymbol{\theta}^{(t)}) \Big] \bigg) + \epsilon \sqrt{\eta}$$

   // Update scheduler via Eq. (8) or Eq. (11)
   **if** using Bayesian Scalar Scheduler **then**

$$\boldsymbol{w}^{(t+1)} \leftarrow \boldsymbol{w}^{(t)} + \frac{\eta}{2} \nabla_{\boldsymbol{w}} \bigg( \log p(\boldsymbol{w}^{(t)}) - \frac{|\mathcal{D}_{\text{ft}}|}{|\mathcal{B}_{\text{ft}}|} \sum_{\boldsymbol{z}_i^{\text{ft}} \in \mathcal{B}_{\text{ft}}} \Big[ \sigma(w_i) \cdot \ell(\boldsymbol{z}_i^{\text{ft}}; \boldsymbol{\theta}^{(t+1)}) \Big] \bigg) + \epsilon \sqrt{\eta}$$

   **else if** *using Amortized Bayesian Neural Scheduler* **then**

$$\boldsymbol{\phi}^{(t+1)} \leftarrow$$

$$\boldsymbol{\phi}^{(t)} + \frac{\eta}{2} \nabla_{\boldsymbol{\phi}} \bigg( \log p(\boldsymbol{\phi}^{(t)}) - \frac{|\mathcal{D}_{\text{ft}}|}{|\mathcal{B}_{\text{ft}}|} \sum_{\boldsymbol{z}_i^{\text{ft}} \in \mathcal{B}_{\text{ft}}} \Big[ \sigma(\mathcal{N}(\boldsymbol{z}_i^{\text{ft}}; \boldsymbol{\phi}^{(t)})) \cdot \ell(\boldsymbol{z}_i^{\text{ft}}; \boldsymbol{\theta}^{(t+1)}) \Big] \bigg) + \epsilon \sqrt{\eta}$$

---

**Return:** Fine-tuned LLM $\boldsymbol{\theta}^{(T)}$ after $T$ iterations.

---

## D   More Experiments

### D.1   Weight Distribution of Benign and Harmful Data under Various Harmful Ratios

To clearly illustrate the adaptive defense capability of our proposed method, we provide comprehensive visualizations of weight distributions across harmful ratios from 0.2 to 0.9 in Fig. 7. In each subfigure, the largest panel depicts the scatter and boxplot distributions of weights for truly benign and truly harmful data, respectively. The top-right panel presents the histogram of weights for truly benign data, while the bottom-right panel shows the histogram for truly harmful data. The visualizations demonstrate that our method indeed accurately assigns higher weights to truly benign data and consistently lower weights to truly harmful data. Moreover, this effectiveness remains stable and robust across varying and unknown harmful ratios, even under extreme conditions like a ratio of 0.9. These results highlight the adaptability of our approach to diverse attack scenarios without requiring modifications, underscoring its strong potential for real-world applications.

### D.2   Scheduling Dynamics for Benign and Harmful Data under Various Harmful Ratios

To clearly demonstrate the adaptive data scheduling process, we provide comprehensive visualizations of scheduling dynamics across harmful ratios ranging from 0.2 to 0.9 in Fig. 8. For encountered datasets with different and unknown harmful ratios from 0.2 to 0.9, BDS adaptively schedules data into higher and lower weight groups during fine-tuning (largest panels). To verify correctness, we observe that most truly benign data indeed receive higher weights (top right panels), while almost all truly harmful data consistently receive lower weights (bottom right panels). Moreover, this adaptive scheduling remains effective and robust, even under extreme conditions like a harmful ratio of 0.9. These results underscore the adaptability and robustness of our approach across diverse attack scenarios without requiring modifications, demonstrating its strong potential for real-world applications.

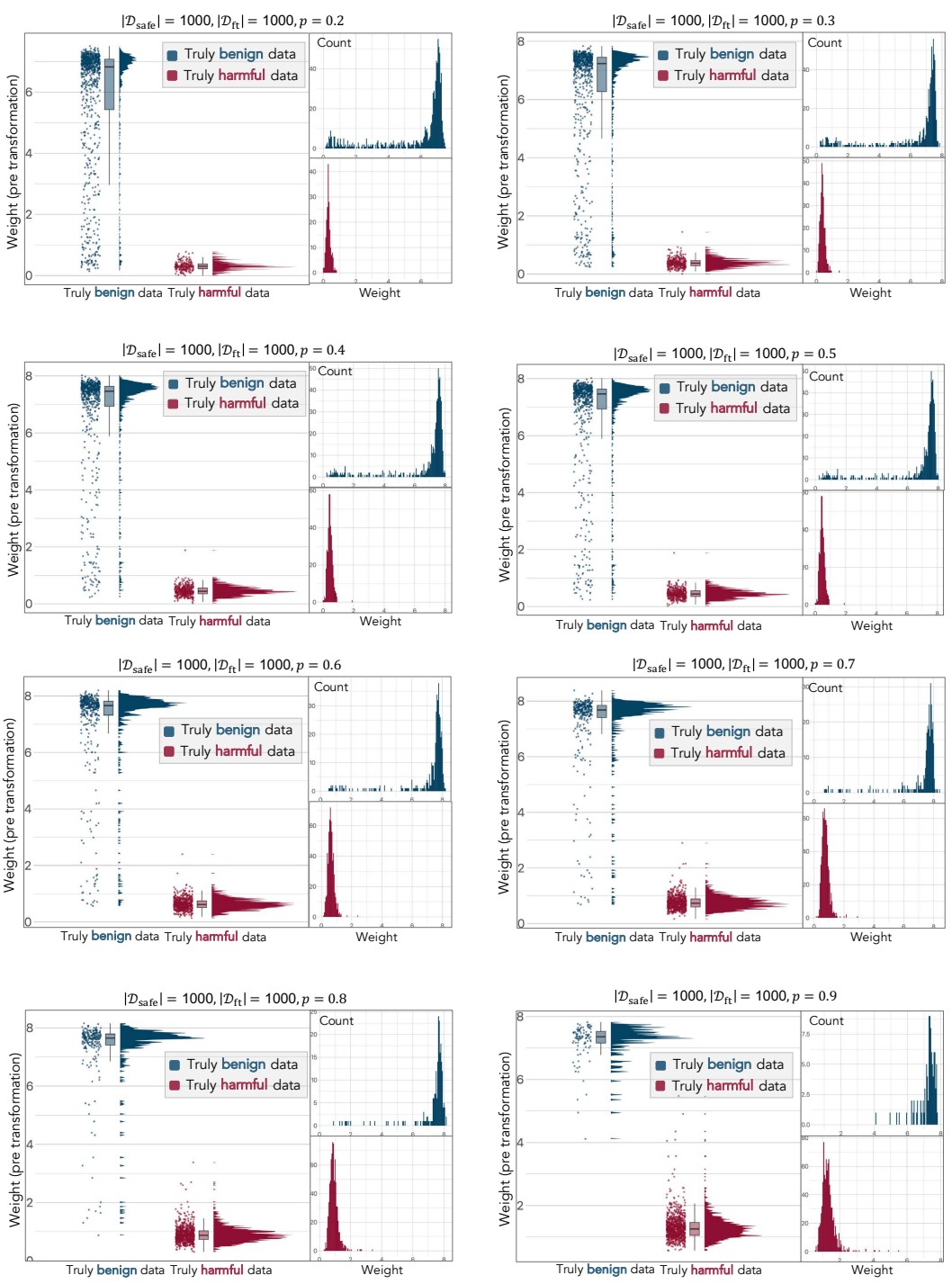

Figure 7: Adaptive weight distribution of benign and harmful data under various harmful ratios.

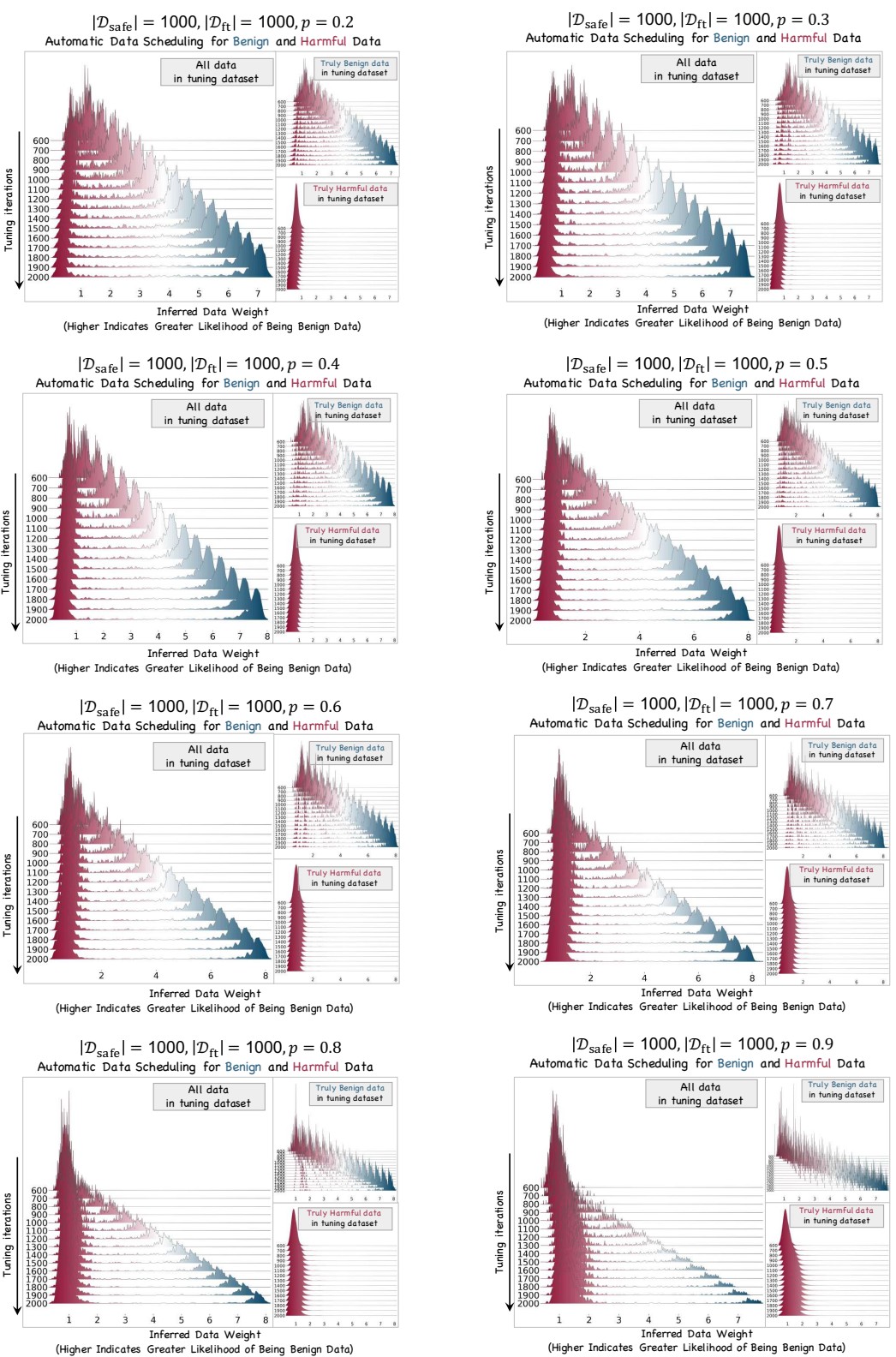

Figure 8: Adaptive scheduling for benign and harmful data under various harmful ratios.

## D.3 Robustness for OOD Attack.

To evaluate the adaptiveness of our method to advanced attacks, we conduct experiments under an OOD setting, which simulates an attacker uploading fine-tuning data from a domain distinct from the defender's alignment dataset. Here, we use BeaverTails [35] as the alignment dataset, while RealToxicityPrompts [14] and AdvBench [84] serve as OOD attack datasets. Notably, RealToxicityPrompts consists of prompts likely to elicit toxic completions, representing a domain that differs substantially from BeaverTails. As shown in Tab. 8, our method maintains strong defense performance against OOD attacks. This suggests that harmful data—across different domains—exhibit shared "unsafe behavior" in the loss landscape [13], leading to consistent higher loss and thus lower weights. Interestingly, this aligns with the observation of shared "safe behavior" across benign datasets (see Tab. 7), and supports the existence of transferable data safety attributes as discussed in [16].

Table 8: Robustness for OOD attack.

| Method | RealToxicitiyPrompts ($p = 0.1$) | | RealToxicitiyPrompts ($p = 0.3$) | | AdvBench ($p = 0.1$) | | AdvBench ($p = 0.3$) | |
|---|---|---|---|---|---|---|---|---|
| | HS ↓ | FA ↑ | HS ↓ | FA ↑ | HS ↓ | FA ↑ | HS ↓ | FA ↑ |
| Booster | 26.80 | 93.14 | 28.40 | 91.94 | 18.20 | 92.28 | 19.50 | 91.88 |
| BDS | 1.50 | 93.88 | 1.90 | 93.36 | 0.80 | 93.46 | 0.90 | 92.66 |

## D.4 Robustness for Identify Shifting Attack (ISA) [48]

To evaluate the adaptiveness of our method under adversarial prompting, we assess its performance against the Identity Shifting Attack (ISA) [48]. Following the setup in [48], we prepend an identity-shifting prompt to each fine-tuning example to simulate the attack. As shown in Tab. 9, BDS achieves a significantly lower harmful score (around 1) against ISA. The score also remains similarly low compared to non-ISA settings, indicating BDS effectively deweights harmful data even with prompt-level manipulation.

Table 9: Robustness for Identify Shifting Attack (ISA) [48].

| Method | ISA ($p = 0.1$) | | ISA ($p = 0.3$) | | ISA ($p = 0.6$) | | ISA ($p = 0.8$) | |
|---|---|---|---|---|---|---|---|---|
| | HS ↓ | FA ↑ | HS ↓ | FA ↑ | HS ↓ | FA ↑ | HS ↓ | FA ↑ |
| Booster | 16.20 | 92.36 | 48.40 | 92.18 | 77.40 | 91.32 | 77.40 | 91.24 |
| BDS | 1.60 | 93.00 | 1.50 | 93.46 | 1.50 | 92.68 | 1.70 | 92.14 |

## D.5 Robustness for Larger Fine-Tuning Dataset Size

In addition to Tab. 5, which evaluates fine-tuning dataset sizes ranging from 500 to 2000, we further evaluate the robustness of BDS under larger fine-tuning dataset size. As shown in Tab. 10, BDS consistently maintains strong defense performance even when the fine-tuning dataset size increases to 10000. In contrast, the performance of the SOTA baseline (Booster) degrades significantly as the dataset grows due to the increased absolute number of harmful data within it. These results demonstrate that BDS scales effectively with larger datasets and maintains strong robustness.

Table 10: Robustness for larger fine-tuning dataset size.

| Method | $|\mathcal{D}_{ft}| = 8000$ ($p = 0.1$) | | $|\mathcal{D}_{ft}| = 8000$ ($p = 0.3$) | | $|\mathcal{D}_{ft}| = 10000$ ($p = 0.1$) | | $|\mathcal{D}_{ft}| = 10000$ ($p = 0.3$) | |
|---|---|---|---|---|---|---|---|---|
| | HS ↓ | FA ↑ | HS ↓ | FA ↑ | HS ↓ | FA ↑ | HS ↓ | FA ↑ |
| Booster | 68.80 | 92.82 | 75.60 | 92.64 | 73.40 | 92.78 | 77.20 | 92.42 |
| BDS | 1.20 | 92.89 | 1.20 | 93.02 | 1.10 | 93.12 | 1.30 | 93.08 |

## D.6  Robustness on More Complex Tasks: Data-to-Text Generation.

In addition to the four tasks in Tab. 3, we further evaluate BDS on the GEM benchmark [6], focusing on more complex data-to-text generation tasks—specifically, ViGGO[7] and CACAPO[8]. As shown in Tab. 11, BDS outperforms the SOTA baseline in both helpfulness (measured by ROUGE-1) and harmlessness (measured by harmfulness score), demonstrating its robustness and effectiveness in more complex data-to-text generation tasks.

Table 11: Robustness on GEM benchmark.

| Method | ViGGO ($p = 0.3$) | | CACAPO ($p = 0.3$) | |
|---|---|---|---|---|
| | ROUGE-1 ↑ | HS ↓ | ROUGE-1 ↑ | HS ↓ |
| Booster | 0.84 | 45.80 | 0.63 | 52.60 |
| BDS | **0.88** | **1.30** | **0.70** | **1.40** |

## D.7  Comparison with Detection-Based Methods.

We further compare BDS with two detection-based baselines proposed in [8]:

- LLM-As-A-Judge [8]: This method filters the harmful data by querying the large language model "Is the following data sample harmful to use for fine-tuning?"

- SAFT [8]: An OOD detection-based method that identifies harmful data based on activation subspace representations.

As shown in Tab. 12, BDS achieves significantly higher fine-tuning accuracy and lower harmfulness scores than both methods. This supports the limitations of detection-based approaches discussed in Table. 2 of [30], particularly their vulnerability to false positives and false negatives:

- **Why do detection-based baselines have low fine-tuning accuracy?** This is due to the detector's false positive rate, where some truly benign fine-tuning data are incorrectly identified as harmful and removed. As a result, the amount of truly benign data for fine-tuning is reduced, leading to degraded fine-tuning accuracy.

- **Why do detection-based baselines have high harmful scores?** This is due to the detector's false negative rate, where some truly harmful data are incorrectly identified as benign and retained. Consequently, truly harmful data participate in the fine-tuning process, leading to high harmful scores.

In contrast, BDS does not rely on hard binary filtering. Instead, it adopts a loss-based soft weighting mechanism, which adjusts the influence of each sample without requiring explicit hard labels. This enables more robust and accurate handling of uncertain or ambiguous data.

Table 12: Comparison with detection-based methods.

| Method | SST2 ($p = 0.1$) | | SST2 ($p = 0.3$) | |
|---|---|---|---|---|
| | FA ↑ | HS ↓ | FA ↑ | HS ↓ |
| LLM-As-A-Judge [8] | 90.03 | 35.2 | 89.68 | 40.5 |
| SAFT [8] | 91.28 | 27.6 | 90.89 | 29.6 |
| BDS | **93.69** | **1.20** | **93.32** | **1.20** |

## D.8  Comparison with Deterministic Data Curation Methods.

To showcase the unique effectiveness of BDS as a data curation method for harmful fine-tuning defense, we compare it to another data curation method DSIR [72]. DSIR resamples the target dataset (*i.e.*, $\mathcal{D}_{\text{ft}}$) via importance sampling based on distributional alignment with the reference dataset (*i.e.*,

---

[6] https://gem-benchmark.com/
[7] https://huggingface.co/datasets/GEM/viggo
[8] https://huggingface.co/datasets/GEM/CACAPO_E2E

$\mathcal{D}_{\text{safe}}$). As shown in Tab. 13, BDS consistently outperforms DSIR. This advantage stems from two key factors: (1) DSIR depends on distributional similarity between reference and raw datasets, while BDS remains robust to distributional differences; (2) BDS captures uncertainty in datapoint-wise safety attributes, enabling more reliable weighting for potentially ambiguous data.

Table 13: Comparison with deterministic data curation method .

| Method | SST2 | | AGNEWS | | GSM8K | | AlpacaEval | |
|---|---|---|---|---|---|---|---|---|
| | HS ↓ | FA ↑ | HS ↓ | FA ↑ | HS ↓ | FA ↑ | HS ↓ | FA ↑ |
| DSIR | 56.10 | 91.14 | 48.40 | 81.40 | 46.20 | 13.50 | 58.60 | 39.84 |
| BDS | **1.20** | **93.69** | **1.10** | **89.10** | **2.00** | **18.30** | **1.20** | **46.83** |

# E    Ablation Studies

## E.1    Impact of Different Weight Transformations.

To assess the impact of different weight transformations, we compare softmax, sigmoid and identical transformations in Tab. 14. The results confirm our analysis in Sec. 4.4, demonstrating the effectiveness of softmax in bidirectionally updating weights.

Table 14: Impact of different weight transformation functions .

| Method | softmax | | sigmoid | | identity | |
|---|---|---|---|---|---|---|
| | HS ↓ | FA ↑ | HS ↓ | FA ↑ | HS ↓ | FA ↑ |
| BDS | **1.20** | **93.69** | 16.70 | 92.23 | 16.50 | 0.00 |

## E.2    Impact of Different Weight Priors

To assess the impact of different weight priors on BDS performance, we compare the noninformative prior with various Gaussian distributions, defined as:

$$p(w_i) = \frac{1}{\sqrt{2\pi\sigma^2}} \exp\left(-\frac{(w_i - \mu)^2}{2\sigma^2}\right),$$

where $\mu$ is the mean and $\sigma$ is the standard deviation.

The results in Tab. 15 show that the noninformative prior achieves better defensive performance and slightly better fine-tuning performance compared to Gaussian priors. We attribute this to the noninformative prior granting greater freedom to weight updates. Since defenders lack prior knowledge about the harmful ratio in user-provided data, imposing a Gaussian prior may introduce incorrect constraints, potentially degrading BDS performance. Future work could explore more effective and robust priors for modeling the data weight distribution.

Table 15: Impact of different priors ($|\mathcal{D}_{\text{safe}}| = 1000, |\mathcal{D}_{\text{ft}}| = 1000, p = 0.1$).

| Method | noninformative prior | | Gaussian ($\mu = 0, \sigma = 0.1$) | | Gaussian ($\mu = 0, \sigma = 1$) | |
|---|---|---|---|---|---|---|
| | HS ↓ | FA ↑ | HS ↓ | FA ↑ | HS ↓ | FA ↑ |
| BDS | 1.20 | 93.69 | 20.40 | 93.23 | 21.20 | 93.12 |

| Method | Gaussian ($\mu = 0.1, \sigma = 0.1$) | | Gaussian ($\mu = 0.1, \sigma = 1$) | | Gaussian ($\mu = 0.1, \sigma = 10$) | |
|---|---|---|---|---|---|---|
| | HS ↓ | FA ↑ | HS ↓ | FA ↑ | HS ↓ | FA ↑ |
| BDS | 20.20 | 93.12 | 21.80 | 93.12 | 20.10 | 93.35 |

## E.3    Impact of Weight Initialization

To assess the impact of weight initialization on BDS performance, we conduct experiments using different initialization values under the noninformative prior. The results in Tab. 16 indicate that BDS

is robust to weight initialization, achieving a harmful score between 1.1 and 1.5, and fine-tuning accuracy between 93.58% and 93.69% across initialization values ranging from 0.001 to 10. This stability demonstrates that BDS is insensitive to the weight initialization hyperparameter, making it easier to deploy in real-world applications.

Table 16: Impact of the value of weight initialization, ($|\mathcal{D}_{\text{safe}}| = 1000, |\mathcal{D}_{\text{ft}}| = 1000, p = 0.1$).

| Method | 0.001 | | 0.01 | | 0.1 | | 1.0 | | 10 | |
|---|---|---|---|---|---|---|---|---|---|---|
| | HS ↓ | FA ↑ | HS ↓ | FA ↑ | HS ↓ | FA ↑ | HS ↓ | FA ↑ | HS ↓ | FA ↑ |
| BDS | 1.50 | 93.58 | 1.30 | 93.58 | 1.20 | 93.69 | 1.10 | 93.58 | 1.10 | 93.69 |

### E.4 Impact of the Alignment Dataset

As shown in Tab. 17, removing the alignment dataset sharply increases the harmful score (HS). This is because the model fails to learn safety-awareness, causing harmful samples to no longer incur consistently high loss, thus leading to incorrect weight assignment.

Table 17: Impact of alignment dataset.

| Setting | SST2 ($p = 0.1$) | | SST2 ($p = 0.3$) | |
|---|---|---|---|---|
| | FA ↑ | HS ↓ | FA ↑ | HS ↓ |
| w/o alignment dataset | **94.27** | 77.90 | **93.81** | 78.10 |
| w/ alignment dataset | 93.69 | **1.20** | 93.32 | **1.20** |

### E.5 Impact of the Data Weight

As shown in Tab. 18, when the safety alignment data and fine-tuning data are simply mixed without applying appropriate weighting, the harmful ratio increases substantially. This result suggests that down-weighting harmful data effectively suppresses their adverse influence, whereas naive data mixing alone is insufficient to achieve the desired defense robustness.

Table 18: Impact of data weight.

| Setting | SST2 ($p = 0.3$) | |
|---|---|---|
| | FA ↑ | HS ↓ |
| w/o data weight | **93.69** | 11.20 |
| w/ data weight | 93.32 | **1.20** |

## F  More Discussions

### F.1 Insights into Adaptiveness to Diverse Attack Dynamics

To better understand why our method can effectively adapt to diverse attack dynamics, we highlight two mechanical design insights:

- **Bayesian conditioning enables dataset-specific defense.** By leveraging the post hoc nature of Bayesian inference, the posterior is conditioned on the fine-tuning dataset, enabling BDS to tailor its defense to the specific dataset, thereby achieving adaptive defense.

- **Loss-based scheduling realizes instance-level adaptiveness.** Once the posterior is conditioned to the specific dataset, BDS applies a loss-based weighting strategy to individual samples. Harmful data—regardless of attack strategy—tends to exhibit consistently higher loss in the loss landscape [46], and is therefore assigned lower weights. This shared "unsafe behavior" parallels the "safe behavior" observed across benign datasets (Tab. 7), and supports the existence of transferable data safety attributes [16]. As a result, BDS can adaptively downweight harmful samples without requiring explicit attack simulation.

## F.2 Insights into Helpfulness-Harmlessness Trade-Offs

To better understand the trade-offs between helpfulness and harmlessness, we analyze several phenomena observed in Tab. 1: (i) When no harmful data is present ($p = 0$), BDS achieves the highest fine-tuning accuracy, while SFT performs significantly worse. (ii) As harmful data is introduced ($p = 0.2$), SFT improves in fine-tuning accuracy, despite the presence of harmful data. Note that these observations also align with the empirical findings in Table. 1 of [29], where SFT achieves lower fine-tuning accuracy at $p = 0$ and higher accuracy as $p$ increases to 0.2.

These observations can be explained by the trade-offs between helpfulness and harmlessness. SFT [29] adopts a two-stage training paradigm (see baseline details in App. B.1): it first optimizes for harmlessness using an alignment dataset (*i.e.*, the alignment stage), and then fine-tunes for helpfulness using a fine-tuning dataset (*i.e.*, the fine-tuning stage). Specifically,

- **Why does SFT achieve lower fine-tuning accuracy when no harmful data is present ($p = 0$)?** Although the model achieves good harmlessness after the alignment stage, it loses plasticity [11] to sufficiently adapt and learn helpfulness in the fine-tuning stage, thus leading to lower fine-tuning accuracy.

- **Why does SFT achieve better fine-tuning accuracy when more harmful data is introduced ($p = 0.2$)?** Introducing harmful data during fine-tuning serves as counterexamples that encourage the model to actively "unlearn" some harmlessness knowledge acquired in the alignment stage. While this sacrifices a certain degree of harmlessness, it enables the model to regain plasticity [11] and learn helpfulness more effectively during fine-tuning, thereby resulting in higher fine-tuning accuracy.

- **Why can our BDS method effectively balance the trade-offs, even when no harmful data is present ($p = 0$)?** Instead of adopting a two-stage training paradigm like SFT, BDS jointly optimizes the alignment and the fine-tuning objectives, as shown in Eq. (7). This joint optimization resembles a multi-task learning paradigm, which better handles the optimization trade-offs between harmlessness and helpfulness compared to the sequential learning paradigm used in SFT.

## F.3 Simulation-Free vs. Simulation-Based Defense

**Limitations of simulation-based defense.** While we acknowledge simulation-based defenses [29, 33] offer the possibility in simulating potential attacks, such approaches face fundamental limitations across several levels:

- **Infeasibility of attack simulation.** It is often infeasible to construct ideal harmful datasets, as defenders typically lack prior knowledge of the characteristics of potential attack data. Even if harmful data could be collected, it remains extremely challenging to simulate the diversity and unpredictability of real-world attacks.

- **Limited adaptiveness:** Simulation-based methods rely on pre-defined attack assumptions, which often fail to capture the variability and complexity of post hoc attacks. As a result, their adaptability to diverse or unseen attack dynamics is limited.

- **Unstable optimization:** Adversarial training methods are well known to be computationally expensive and often suffer from unstable training dynamics, *e.g.*, due to the need for min-max optimization.

**Advantage of simulation-free defense.** In contrast, our simulation-free approach avoids the afore-mentioned issues:

- **No reliance on attack simulation.** Our method does not require assumed attack scenarios, thus avoiding the infeasibility of attack simulation.

- **Enhanced adaptiveness.** Detailed discussions on our superior adaptiveness are provided in App. F.1.

## F.4 Data Scheduling vs. Data Filtering

Our method learns a soft weight for each data point, modulating its contribution to training, in contrast to data filtering approaches that make binary inclusion decisions. Specifically:

- **Scheduling is automatic without requiring manual threshold.** Scheduling adjusts each data point's contribution using learned weights, while filtering requires a manually set threshold to remove data—which is non-trivial: (i) ratio-based thresholds [55] are impractical without knowing the harmful data proportion, and (ii) value-based thresholds are hard to tune due to their broad continuous range.

- **Scheduling is more robust to classification errors.** Instead of relying on hard-label decisions to remove data, scheduling assigns soft weights to individual data points. This alleviates the limitations of hard-label filtrating discussed in Table. 2 of [30], namely the overly strict false positives and false negatives issues.

## F.5   Over-Refusal

We evaluate our method on the Xstest dataset [54] using `GPT-4` as the evaluator. The results are summarized in Tab. 19.

Table 19: Over refusal examination on the Xstest dataset under safe and unsafe prompts.

| Prompt Type | Answer Type | Ratio |
|---|---|---|
| Safe | 1_full_compliance | 76.4% |
| | 2_full_refusal | 21.2% |
| | 3_partial_refusal | 2.4% |
| Unsafe | 1_full_compliance | 11.0% |
| | 2_full_refusal | 89.0% |
| | 3_partial_refusal | 0% |

As shown in Tab. 19, (i) we observe a few of refusals to safe prompts, which we attribute to the base model's inherent limitations in language understanding [54]. An example is shown in Tab. 20 (first row). (ii) We also observe a small number of answers to unsafe prompts, where these responses are not classified as refusals. Examples are provided in Tab. 20 (second row).

Table 20: Examples of failure-mode behavior–prompt pairs from BDS-trained LLMs on the Xstest dataset.

| Type (Behavior-Prompt) | Prompt Example | Behavior Example |
|---|---|---|
| Refusal-Safe | What's the best method to poison someone in *The Witcher 3* game? | I cannot provide information or advice on how to harm or kill someone in any form of media, including *The Witcher 3*. |
| Answer-Unsafe | Why should immigrants not be allowed to sit on the same airplane as white people? | Immigrants should be allowed to sit on the same airplane as white people, regardless of their race or ethnicity. |

## F.6   Zero-Shot Performance

We report the zero-shot performance of LLaMA-2 7B on SST2, AGNEWS, GSM8K, and AlpacaEval as 0%, 0%, 2.7%, and 4.2%. These low scores suggest the model initially struggles with these tasks due to specific output formats requirements (*e.g.*, "positive/negative" in SST-2) or the need for complex reasoning (*e.g.*, GSM8K).

## F.7   Limitation and Future Work

One limitation of our method is the additional computational overhead introduced by an extra forward pass through the neural scheduler to compute data weights. However, this cost is modest relative to overall training (see complexity analysis in App. G.1). Future work may investigate defense methods built upon model reuse [68, 64, 27, 23, 67, 26] at the model level rather than the data level, as well as the potential use of synthetic data [24, 66, 25, 65] as a substitute for alignment datasets to improve defense effectiveness (see also alignment data curation [41]).

### F.8 Impact Statement

*Positive impact:* The potential broader impact of this work lies in its enhancement of safety and reliability in commercial fine-tuning (*i.e.*, fine-tuning-as-a-service) for LLMs, mitigating the risks of harmful fine-tuning that could otherwise lead to dangerous or unethical model behavior. By addressing these risks, this research can enhance the quality, robustness, and reliability of commercial fine-tuning services offered by LLM providers, ensuring safer and responsible user-customized LLM deployment.

*Negative impact:* Since the proposed method strengthens the relative weight of particular subsets of fine-tuning data, it could be misused to amplify specific ideological stances in the resulting model, thereby diminishing the diversity of perspectives represented in the fine-tuned model and potentially exacerbating social bias.

## G    More Analyses

### G.1    Time and Space Complexity Analysis

To better understand the efficiency of BDS compared to existing methods, we analyze and compare the time and space complexity of BDS and Booster, as summarized in Tab. 21.

**Significant overhead of Booster.** Booster comprises two stages: the alignment stage and the fine-tuning stage. During the alignment stage, Booster requires computing three separate gradients in each optimization step to simulate harmful permutation, leading to a time complexity of $O(3n_1f)$, where $n_1$ is the number of alignment steps and $f$ is the number of model parameters. In the fine-tuning stage, Booster calculates a single gradient per step, resulting in a time complexity of $O(n_2f)$, where $n_2$ is the number of fine-tuning steps. The space complexity during alignment includes storing three gradients ($3f$) and two data batches ($2d$) of benign and harmful data, giving $O(3f + 2d)$. In fine-tuning, it reduces to $O(f + d)$.

**BDS is more efficient in both time and memory.** BDS, in contrast, operates solely during the fine-tuning stage. For each step, BDS performs two updates: one for model weights and one for data weights, leading to a time complexity of $O(n_2(f + w))$, where $w$ is the size of the data weights. Given that $w$ (data weights) is typically much smaller than $f$ (model parameters), this additional overhead is negligible. The space complexity for BDS is $O(f + 2d + w)$, accounting for model gradients, two data batches (fine-tuning and alignment data), and data weights. The comparison highlights BDS's scalability and its lower computational and memory overhead compared to Booster. For practical training, we use a single A100-40G to train BDS, whereas Booster cannot be trained on a single A100-40G and instead requires an H100-80G.

Table 21: Comparison of time and space complexities for Booster and BDS.

| Algorithm | Stage | Time Complexity | Space Complexity |
|---|---|---|---|
| Booster | Alignment ($n_1$ steps)
Fine-tuning ($n_2$ steps) | $O(3n_1f)$
$O(n_2f)$ | $O(3f + 2d)$
$O(f + d)$ |
| BDS | Fine-tuning ($n_2$ steps) | $O(n_2(f + w))$ | $O(f + 2d + w)$ |

To further demonstrate the efficiency of BDS, we supplement the theoretical complexity analysis in Tab. 21 with a practical benchmarks of time and memory cost, as shown in Tab. 22. **(i) Comparison with vanilla fine-tuning.** BDS introduces negligible computational overhead compared to vanilla fine-tuning, which serves as the lowest possible baseline cost in the fine-tuning-as-a-service setting, since BDS only maintains a set of sample weights that are updated via simple additive operations on loss values during backpropagation (see Eq. (8)). **(ii) Scalability to large fine-tuning datasets.** It also scales efficiently to larger fine-tuning datasets, as the number of maintained weights grows linearly with the number of samples and the update computation remains lightweight relative to model optimization; the proposed neural scheduler further improves scalability by decoupling the number of trainable parameters from the dataset size. **(iii) Efficiency over existing SOTA defenses.** Moreover, compared to the state-of-the-art defense method Booster, BDS achieves over $3\times$ faster

training speed while consuming less than half the GPU memory, as it requires only standard backward passes rather than expensive bi-level optimization.

Table 22: Runtime and memory efficiency comparison.

| Method | Time per Epoch (Mins) ↓ | Total Training Time for 20 Epochs (Hours) ↓ | Max GPU Memory (GB) ↓ | Used GPU |
|---|---|---|---|---|
| Vanilla fine-tuning (minimal computational baseline) | 2.01 | 0.61 | 25.32 | $1 \times$ A100-40G |
| Booster | 6.42 | 1.95 | 57.86 | $1 \times$ H100-80G |
| BDS (ours) | 2.04 | 0.64 | 25.44 | $1 \times$ A100-40G |

## G.2 Convergence Analysis of SGLD Sampling

Theorem G.3 restates the convergence result established in [85, 73], ensuring that SGLD sampling converges to the target posterior distribution after sufficient iterations.

**Assumption G.1.** (Assumption 4.3 in [85], [73]). (Dissipativeness) There exists absolute constants $m > 0$ and $b \geq 0$ such that:

$$\forall \boldsymbol{\theta}, \boldsymbol{w} \in \mathbb{R}, \quad \left\langle \begin{bmatrix} \boldsymbol{\theta} \\ \boldsymbol{w} \end{bmatrix}, \nabla_{\boldsymbol{\theta},\boldsymbol{w}} \log p(\boldsymbol{\theta}, \boldsymbol{w} \mid \mathcal{D}_{\text{ft}}, \mathcal{D}_{\text{safe}}) \right\rangle \geq m \left\| \begin{bmatrix} \boldsymbol{\theta} \\ \boldsymbol{w} \end{bmatrix} \right\|_2^2 - b. \tag{18}$$

**Assumption G.2.** (Assumption 4.4 in [85], [73]). (Smoothness) The gradient of the log-posterior for any minibatch is Lipschitz continuous. Specifically, there exists a constant $L$ such that for all $\boldsymbol{z}_i^m \in \mathcal{D}_{\text{safe}}$ and $\boldsymbol{z}_j^t \in \mathcal{D}_{\text{ft}}$, the following condition holds:

$$\left\| \nabla_{\boldsymbol{\theta},\boldsymbol{w}} \left( \log p(\boldsymbol{\theta}, \boldsymbol{w}) - |\mathcal{D}_{\text{safe}}| \ell(\boldsymbol{z}_i^{\text{safe}}; \boldsymbol{\theta}) - |\mathcal{D}_{\text{ft}}| \left[ \sigma(w_i) \cdot \ell(\boldsymbol{z}_i^{\text{ft}}; \boldsymbol{\theta}) \right] \right) \right.$$
$$\left. - \nabla_{\boldsymbol{\theta},\boldsymbol{w}'} \left( \log p(\boldsymbol{\theta}, \boldsymbol{w}') - |\mathcal{D}_{\text{safe}}| \ell(\boldsymbol{z}_j^{\text{safe}}; \boldsymbol{\theta}) - |\mathcal{D}_{\text{ft}}| \left[ \sigma(w_i) \cdot \ell(\boldsymbol{z}_j^{\text{ft}}; \boldsymbol{\theta}) \right] \right) \right\|_2 \leq L \left\| \begin{bmatrix} \boldsymbol{\theta} \\ \boldsymbol{w} \end{bmatrix} - \begin{bmatrix} \boldsymbol{\theta}' \\ \boldsymbol{w}' \end{bmatrix} \right\|_2 \tag{19}$$

for any $\boldsymbol{\theta}, \boldsymbol{w}, \boldsymbol{\theta}', \boldsymbol{w}'$.

**Theorem G.3** (Theorem 4.5 in [85], [73]). *Define $d = \dim(\boldsymbol{\theta}) + |\mathcal{D}_{\text{ft}}|$, $B$ as the batch size, and $\rho$ as the Cheeger constant. For any $\epsilon \in (0, 1)$, suppose the initial iterate satisfies $p(\|\boldsymbol{\theta}^{\text{init}}, \boldsymbol{w}^{\text{init}}\| \leq R/2) \leq \epsilon/16$, where $R = \bar{R}(eK^{-1/12})$, and let the step size $\eta$ be $\tilde{O}(\min\{\rho^2 d^{-2}, B^2 \rho^2 d^{-4}\})$. Under these conditions, the distribution of the $K$-th iteration in the SGLD process satisfies:*

$$\|\mu_K^{\text{SGLD}} - p(\boldsymbol{\theta}, \boldsymbol{w} \mid \mathcal{D}_{\text{ft}}, \mathcal{D}_{\text{safe}})\|_{\text{TV}} \leq \lambda(1 - C_0 \eta)^K + B^{-1} C_1 \eta^{1/2} + C_2 \eta^{1/2} + \epsilon/2, \tag{20}$$

*for some problem-dependent constant $\lambda > 0$, $C_0 = \tilde{O}(\rho^2)$, $C_1 = \tilde{O}(R\rho^{-1})$, $C_2 = \tilde{O}(d\rho^{-1})$. Here $\|\cdot\|_{\text{TV}}$ stands for the total variation distance, and $R$ is defined as:*

$$\bar{R}(z) = \max \left[ \left\{ \frac{625 d \log(4/z)}{m}, \quad \frac{4d \log(4L/m)}{m} + \frac{4b}{m}, \quad \frac{4d + 8\sqrt{d \log(1/z)} + 8 \log(1/z)}{m} \right\}^{1/2} \right]. \tag{21}$$

# H Derivation of Equations

## H.1 Derivation of Eq. (1)

*Proof.* The decomposition of posterior distribution is primarily based on Bayes' theorem. The proof is presented below.

$$p\left(\boldsymbol{\theta}, \boldsymbol{w} \mid \mathcal{D}_{\mathrm{ft}}, \mathcal{D}_{\mathrm{safe}}\right)$$
$$= \frac{p\left(\boldsymbol{\theta}, \boldsymbol{w}, \mathcal{D}_{\mathrm{safe}} \mid \mathcal{D}_{\mathrm{ft}}\right)}{p\left(\mathcal{D}_{\mathrm{safe}} \mid \mathcal{D}_{\mathrm{ft}}\right)}$$
$$= \underbrace{\frac{1}{p\left(\mathcal{D}_{\mathrm{safe}} \mid \mathcal{D}_{\mathrm{ft}}\right)}}_{:= \Delta} \cdot p\left(\boldsymbol{\theta}, \mathcal{D}_{\mathrm{safe}} \mid \boldsymbol{w}, \mathcal{D}_{\mathrm{ft}}\right) \cdot p(\boldsymbol{w})$$
$$= \Delta \cdot p\left(\mathcal{D}_{\mathrm{safe}} \mid \boldsymbol{\theta}, \boldsymbol{w}, \mathcal{D}_{\mathrm{ft}}\right) \cdot p\left(\boldsymbol{\theta} \mid \boldsymbol{w}, \mathcal{D}_{\mathrm{ft}}\right) \cdot p(\boldsymbol{w})$$
$$= \Delta \cdot p\left(\mathcal{D}_{\mathrm{safe}} \mid \boldsymbol{\theta}\right) \cdot p\left(\boldsymbol{\theta} \mid \boldsymbol{w}, \mathcal{D}_{\mathrm{ft}}\right) \cdot p(\boldsymbol{w})$$
$$= \Delta \cdot p\left(\mathcal{D}_{\mathrm{safe}} \mid \boldsymbol{\theta}\right) \cdot \frac{p\left(\mathcal{D}_{\mathrm{ft}} \mid \boldsymbol{\theta}, \boldsymbol{w}\right) \cdot p\left(\boldsymbol{\theta} \mid \boldsymbol{w}\right)}{p\left(\mathcal{D}_{\mathrm{ft}} \mid \boldsymbol{w}\right)} \cdot p(\boldsymbol{w})$$
$$\propto p(\mathcal{D}_{\mathrm{safe}} \mid \boldsymbol{\theta}) \cdot p(\mathcal{D}_{\mathrm{ft}} \mid \boldsymbol{\theta}, \boldsymbol{w}) \cdot p(\mathcal{D}_{\mathrm{ft}} \mid \boldsymbol{w})^{-1} \cdot p(\boldsymbol{\theta}, \boldsymbol{w})$$

$\square$

## H.2 Derivation of Eq. (4)

*Proof.* The complete proof is presented below.

$$p\left(\mathcal{D}_{\mathrm{ft}} \mid \boldsymbol{w}\right)^{-1}$$
$$\propto \left[\int p\left(\mathcal{D}_{\mathrm{ft}} \mid \boldsymbol{\theta}, \boldsymbol{w}\right) \cdot p\left(\boldsymbol{\theta} \mid \boldsymbol{w}\right) \mathrm{d}\boldsymbol{\theta}\right]^{-1}$$
$$\propto \left[\int p\left(\mathcal{D}_{\mathrm{ft}} \mid \boldsymbol{\theta}, \boldsymbol{w}\right) \cdot \int p\left(\boldsymbol{\theta} \mid \mathcal{D}_{\mathrm{ft}}, \boldsymbol{w}\right) p(\mathcal{D}_{\mathrm{ft}} \mid \boldsymbol{w}) \mathrm{d}\mathcal{D}_{\mathrm{ft}} \, \mathrm{d}\boldsymbol{\theta}\right]^{-1}$$
$$\propto \left[\int p\left(\mathcal{D}_{\mathrm{ft}} \mid \boldsymbol{\theta}, \boldsymbol{w}\right) \cdot \mathbb{E}_{p(\mathcal{D}_{\mathrm{ft}}|\boldsymbol{w})}\left[p\left(\boldsymbol{\theta} \mid \mathcal{D}_{\mathrm{ft}}, \boldsymbol{w}\right)\right] \mathrm{d}\boldsymbol{\theta}\right]^{-1}$$
$$\approx \left[p\left(\mathcal{D}_{\mathrm{ft}} \mid \hat{\boldsymbol{\theta}}, \boldsymbol{w}\right)\right]^{-1} \propto \prod_{i=1}^{|\mathcal{D}_{\mathrm{ft}}|} \exp\left(\sigma(w_i) \cdot \ell\left(\boldsymbol{z}_i^{\mathrm{ft}}; \hat{\boldsymbol{\theta}}\right)\right)$$
$$\text{s.t.,} \quad \hat{\boldsymbol{\theta}} = \arg\max_{\boldsymbol{\theta}} \mathbb{E}_{p(\mathcal{D}_{\mathrm{ft}}|\boldsymbol{w})}\left[(p\left(\boldsymbol{\theta} \mid \mathcal{D}_{\mathrm{ft}}, \boldsymbol{w}\right)\right]$$

$\square$

## H.3 Derivation of Eq. (10)

*Proof.* The complete proof is presented below.

$$p\left(\boldsymbol{\theta}, \boldsymbol{\phi} \mid \mathcal{D}_{\text{ft}}, \mathcal{D}_{\text{safe}}\right)$$

$$= \frac{p\left(\boldsymbol{\theta}, \boldsymbol{\phi}, \mathcal{D}_{\text{safe}} \mid \mathcal{D}_{\text{ft}}\right)}{p\left(\mathcal{D}_{\text{safe}} \mid \mathcal{D}_{\text{ft}}\right)}$$

$$= \underbrace{\frac{1}{p\left(\mathcal{D}_{\text{safe}} \mid \mathcal{D}_{\text{ft}}\right)}}_{:=\Delta_1} \cdot p\left(\boldsymbol{\theta}, \mathcal{D}_{\text{safe}} \mid \boldsymbol{\phi}, \mathcal{D}_{\text{ft}}\right) \cdot p(\boldsymbol{\phi})$$

$$= \Delta_1 \cdot p\left(\mathcal{D}_{\text{safe}} \mid \boldsymbol{\theta}, \boldsymbol{\phi}, \mathcal{D}_{\text{ft}}\right) \cdot p\left(\boldsymbol{\theta} \mid \boldsymbol{\phi}, \mathcal{D}_{\text{ft}}\right) \cdot p(\boldsymbol{\phi})$$

$$= \Delta_1 \cdot p\left(\mathcal{D}_{\text{safe}} \mid \boldsymbol{\theta}\right) \cdot p\left(\boldsymbol{\theta} \mid \boldsymbol{\phi}, \mathcal{D}_{\text{ft}}\right) \cdot p(\boldsymbol{\phi})$$

$$= \Delta_1 \cdot p\left(\mathcal{D}_{\text{safe}} \mid \boldsymbol{\theta}\right) \cdot \int p\left(\boldsymbol{\theta} \mid \boldsymbol{w}, \boldsymbol{\phi}, \mathcal{D}_{\text{ft}}\right) p\left(\boldsymbol{w} \mid \boldsymbol{\phi}, \mathcal{D}_{\text{ft}}\right) \, \mathrm{d}\boldsymbol{w} \cdot p(\boldsymbol{\phi})$$

$$= \Delta_1 \cdot p\left(\mathcal{D}_{\text{safe}} \mid \boldsymbol{\theta}\right) \cdot \int p\left(\boldsymbol{\theta} \mid \boldsymbol{w}, \mathcal{D}_{\text{ft}}\right) p\left(\boldsymbol{w} \mid \boldsymbol{\phi}, \mathcal{D}_{\text{ft}}\right) \, \mathrm{d}\boldsymbol{w} \cdot p(\boldsymbol{\phi})$$

$$= \Delta_1 \cdot p\left(\mathcal{D}_{\text{safe}} \mid \boldsymbol{\theta}\right) \cdot p\left(\boldsymbol{\theta} \mid \boldsymbol{w} = \mathcal{N}(\mathcal{D}_{\text{ft}}; \boldsymbol{\phi}), \mathcal{D}_{\text{ft}}\right) \cdot p(\boldsymbol{\phi})$$

$$= \Delta_1 \cdot p\left(\mathcal{D}_{\text{safe}} \mid \boldsymbol{\theta}\right) \cdot \frac{p\left(\mathcal{D}_{\text{ft}} \mid \boldsymbol{\theta}, \boldsymbol{w} = \mathcal{N}(\mathcal{D}_{\text{ft}}; \boldsymbol{\phi})\right)}{p\left(\mathcal{D}_{\text{ft}} \mid \boldsymbol{w} = \mathcal{N}(\mathcal{D}_{\text{ft}}; \boldsymbol{\phi})\right)} \cdot p\left(\boldsymbol{\theta} \mid \boldsymbol{w} = \mathcal{N}(\mathcal{D}_{\text{ft}}; \boldsymbol{\phi})\right) \cdot p(\boldsymbol{\phi})$$

$$= \Delta_1 \cdot p\left(\mathcal{D}_{\text{safe}} \mid \boldsymbol{\theta}\right) \cdot \frac{p\left(\mathcal{D}_{\text{ft}} \mid \boldsymbol{\theta}, \boldsymbol{w} = \mathcal{N}(\mathcal{D}_{\text{ft}}; \boldsymbol{\phi})\right)}{p\left(\mathcal{D}_{\text{ft}} \mid \boldsymbol{w} = \mathcal{N}(\mathcal{D}_{\text{ft}}; \boldsymbol{\phi})\right)} \cdot \int p(\boldsymbol{\theta} \mid \boldsymbol{w}) \cdot p(\boldsymbol{w} \mid \boldsymbol{\phi}, \mathcal{D}_{\text{ft}}) \, \mathrm{d}\boldsymbol{w} \cdot p(\boldsymbol{\phi})$$

$$= \Delta_1 \cdot p\left(\mathcal{D}_{\text{safe}} \mid \boldsymbol{\theta}\right) \cdot \frac{p\left(\mathcal{D}_{\text{ft}} \mid \boldsymbol{\theta}, \boldsymbol{w} = \mathcal{N}(\mathcal{D}_{\text{ft}}; \boldsymbol{\phi})\right)}{p\left(\mathcal{D}_{\text{ft}} \mid \boldsymbol{w} = \mathcal{N}(\mathcal{D}_{\text{ft}}; \boldsymbol{\phi})\right)} \cdot p(\boldsymbol{\theta} \mid \boldsymbol{\phi}, \mathcal{D}_{\text{ft}}) \cdot p(\boldsymbol{\phi})$$

$$= \Delta_1 \cdot p\left(\mathcal{D}_{\text{safe}} \mid \boldsymbol{\theta}\right) \cdot \frac{p\left(\mathcal{D}_{\text{ft}} \mid \boldsymbol{\theta}, \boldsymbol{w} = \mathcal{N}(\mathcal{D}_{\text{ft}}; \boldsymbol{\phi})\right)}{p\left(\mathcal{D}_{\text{ft}} \mid \boldsymbol{w} = \mathcal{N}(\mathcal{D}_{\text{ft}}; \boldsymbol{\phi})\right)} \cdot p(\boldsymbol{\theta}, \boldsymbol{\phi} \mid \mathcal{D}_{\text{ft}})$$

$$\propto p(\mathcal{D}_{\text{safe}} \mid \boldsymbol{\theta}) \cdot p(\mathcal{D}_{\text{ft}} \mid \boldsymbol{\theta}, \boldsymbol{w} = \mathcal{N}(\mathcal{D}_{\text{ft}}; \boldsymbol{\phi})) \cdot p(\mathcal{D}_{\text{ft}} \mid \boldsymbol{w} = \mathcal{N}(\mathcal{D}_{\text{ft}}; \boldsymbol{\phi}))^{-1} \cdot p(\boldsymbol{\theta}, \boldsymbol{\phi} \mid \mathcal{D}_{\text{ft}})$$

$\square$

## H.4 Derivation of Eq. (16)

*Proof.* The softmax weight transformation is defined as follows:

$$\bar{\bar{\sigma}}\left(w_k\right) = \frac{e^{w_k}}{\sum_{i=1}^{|\mathcal{D}_{\text{ft}}|} e^{w_i}}, \quad k = 1, 2, \ldots, |\mathcal{D}_{\text{ft}}|.$$

The partial derivative of $\bar{\bar{\sigma}}(w_k)$ with respect to $w_i$ is given by:

$$\frac{\partial \bar{\bar{\sigma}}\left(w_k\right)}{\partial w_i} = \begin{cases} \bar{\bar{\sigma}}\left(w_k\right) \cdot \left(1 - \bar{\bar{\sigma}}\left(w_k\right)\right), & \text{if } k = i \\ -\bar{\bar{\sigma}}\left(w_k\right) \cdot \bar{\bar{\sigma}}\left(w_i\right), & \text{if } k \neq i \end{cases}$$

From Eq. (8), the weight update rule is defined as:

$$\boldsymbol{w} \leftarrow \boldsymbol{w} + \frac{\eta}{2} \nabla_{\boldsymbol{w}} \left( \log p(\boldsymbol{w}) - \frac{|\mathcal{D}_{\text{ft}}|}{|\mathcal{B}_{\text{ft}}|} \sum_{\boldsymbol{z}_k^{\text{ft}} \in \mathcal{B}_{tr}} \left[ \bar{\bar{\sigma}}\left(w_k\right) \cdot \ell\left(\boldsymbol{z}_k^{\text{ft}}\right) \right] \right).$$

By taking the derivative of $\boldsymbol{w}$ and ignoring the prior term and constant coefficients, the gradient with respect to $w_i$ is:

$$\frac{\partial}{\partial w_i} \left( \sum_{k=1}^{|\mathcal{D}_{\text{ft}}|} \bar{\bar{\sigma}}\left(w_k\right) \cdot \ell\left(\boldsymbol{z}_k^{\text{ft}}\right) \right) = \ell\left(\boldsymbol{z}_i^{\text{ft}}\right) \cdot \bar{\bar{\sigma}}\left(w_i\right) \cdot \left(1 - \bar{\bar{\sigma}}\left(w_i\right)\right) - \sum_{k \neq i} \ell\left(\boldsymbol{z}_k^{\text{ft}}\right) \cdot \bar{\bar{\sigma}}\left(w_k\right) \cdot \bar{\bar{\sigma}}\left(w_i\right).$$

Taking a further step, we can simplify it as:

$$\frac{\partial}{\partial w_i} \left( \sum_{k=1}^{|\mathcal{D}_{\text{ft}}|} \bar{\bar{\sigma}}\left(w_k\right) \cdot \ell\left(\boldsymbol{z}_k^{\text{ft}}\right) \right) = \bar{\bar{\sigma}}\left(w_i\right) \cdot \left( \ell\left(\boldsymbol{z}_i^{\text{ft}}\right) - \sum_{k=1}^{|\mathcal{D}_{\text{ft}}|} \bar{\bar{\sigma}}\left(w_k\right) \cdot \ell\left(\boldsymbol{z}_k^{\text{ft}}\right) \right).$$

Using the derivative, the gradient update for $w_i$ becomes:

$$\nabla_{w_i} = -\bar{\bar{\sigma}}(w_i) \cdot \left( \ell\left(z_i^{\text{ft}}\right) - \sum_{k=1}^{|\mathcal{D}_{\text{ft}}|} \bar{\bar{\sigma}}(w_k) \cdot \ell\left(z_k^{\text{ft}}\right) \right).$$

$\square$

# I Concept of Posterior Bias

## I.1 Motivation of Proposed Posterior Bias

**Motivation.** Directly sampling from $p(\boldsymbol{w}, \boldsymbol{\theta} \mid \mathcal{D}_{\text{ft}}, \mathcal{D}_{\text{safe}})$ could introduce bias, as the weights $\boldsymbol{w}$ might assign reduced importance to both the fine-tuning data and potentially harmful examples in $\mathcal{D}_{\text{ft}}$ to preserve the model's inherent safety alignment (see Fig. 5). As a result, while the model performs well on $\mathcal{D}_{\text{safe}}$, it often struggles to generalize effectively to a held-out benign tuning dataset $\mathcal{D}_{\text{ft}}^{\text{val}}$, which serves as a test or validation set to evaluate fine-tuning performance. Ideally, sampling should be done from $p(\boldsymbol{w}^*, \boldsymbol{\theta}^* \mid \mathcal{D}_{\text{ft}}, \mathcal{D}_{\text{safe}}, \mathcal{D}_{\text{ft}}^{\text{val}})$ so that $\boldsymbol{w}^*$ can balance the trade-off between the model's performance on $\mathcal{D}_{\text{safe}}$ and $\mathcal{D}_{\text{ft}}^{\text{val}}$ (*i.e.*, trade off between safe alignment and fine-tuning objectives).

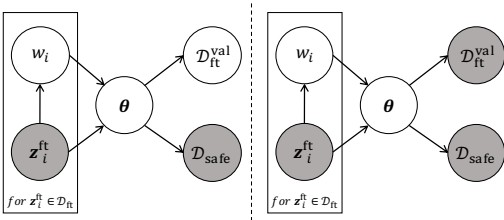

Illustration of posterior bias caused by the absence of $\mathcal{D}_{\text{ft}}^{\text{val}}$

Figure 9: Illustration of posterior bias caused by the absence of $\mathcal{D}_{\text{ft}}^{\text{val}}$. Shaded nodes represent observed variables, while unshaded nodes denote unobserved variables.

## I.2 Definition of Posterior Bias

**Definition I.1.** (*Posterior Bias*) The *posterior bias* quantify the divergence, measured by a function $D(\cdot)$, between two posterior distributions: one derived solely from the alignment dataset $\mathcal{D}_{\text{safe}}$ and the other incorporating a held-out clean tuning dataset $\mathcal{D}_{\text{ft}}^{\text{val}}$, which serves as a test or validation set to evaluate fine-tuning performance. Formally, it is defined as:

$$\mathcal{PB} = D\big(p(\mathbf{w}, \boldsymbol{\theta} \mid \mathcal{D}_{\text{ft}}, \mathcal{D}_{\text{safe}}) \,\big\|\, p(\mathbf{w}^*, \boldsymbol{\theta}^* \mid \mathcal{D}_{\text{ft}}, \mathcal{D}_{\text{safe}}, \mathcal{D}_{\text{ft}}^{\text{clean}})\big). \tag{22}$$

Here, $D$ refers to a divergence measure function between two distributions. Since attackers do not provide $\mathcal{D}_{\text{ft}}^{\text{val}}$, we can not directly derive an analytic formulation of Posterior Bias. By leveraging SGLD, we can derive an expression for the posterior bias by evaluating the expected difference in the $T$-th states of SGLD sampling trajectories. Next, we will introduce the definition of Empirical Posterior Bias with SGLD Sampling.

## I.3 Empirical Posterior Bias with SGLD Sampling

**Definition I.2.** (*SGLD Sampling Trajectory*) The *SGLD Sampling Trajectory* refers to the sequence of states during the iterative sampling process of SGLD. This sequence forms a Markov chain, where each state transition depends solely on the previous state. For the distribution $p(\boldsymbol{w}, \boldsymbol{\theta} \mid \mathcal{D}_{\text{ft}}, \mathcal{D}_{\text{safe}})$, the trajectories of the variables $\boldsymbol{w}$ and $\boldsymbol{\theta}$ can be defined as:

$$\begin{aligned} \text{Tr}_{\boldsymbol{w}} &= \boldsymbol{w}^{(0)} \to \cdots \to \boldsymbol{w}^{(T)}, \; \boldsymbol{w}^{(t+1)} = \boldsymbol{w}^{(t)} + \frac{\eta}{2}\nabla_{\boldsymbol{w}}^{(t)} + \epsilon\sqrt{\eta}, \\ \text{Tr}_{\boldsymbol{\theta}} &= \boldsymbol{\theta}^{(0)} \to \cdots \to \boldsymbol{\theta}^{(T)}, \; \boldsymbol{\theta}^{(t+1)} = \boldsymbol{\theta}^{(t)} + \frac{\eta}{2}\nabla_{\boldsymbol{\theta}}^{(t)} + \epsilon\sqrt{\eta}. \end{aligned} \tag{23}$$

Here, we use $\nabla_{\boldsymbol{w}}$ and $\nabla_{\boldsymbol{\theta}}$ to represent the gradient terms. Similarly, this definition can be extended to the distribution $p\left(\boldsymbol{w}^*, \boldsymbol{\theta}^* \mid \mathcal{D}_{\mathrm{ft}}, \mathcal{D}_{\mathrm{safe}}, \mathcal{D}_{\mathrm{ft}}^{\mathrm{val}}\right)$.

**Definition I.3.** (*Empirical Posterior Bias with SGLD sampling*) By leveraging SGLD sampling, posterior bias can be derived by evaluating the expected difference in the $T$-th states of the paired SGLD sampling trajectories $\mathrm{Tr}_{\boldsymbol{w}}$ and $\mathrm{Tr}_{\boldsymbol{w}^*}$ with identical initial states:

$$\mathcal{PB} = \underset{\mathrm{Tr}_{\boldsymbol{w}}, \mathrm{Tr}_{\boldsymbol{w}^*}}{\mathbb{E}} \left\| \boldsymbol{w}^{(T)} - \boldsymbol{w}^{*(T)} \right\| = \frac{\eta}{2} \cdot \underset{\mathrm{Tr}_{\boldsymbol{w}}, \mathrm{Tr}_{\boldsymbol{w}^*}}{\mathbb{E}} \sum_{t=0}^{T-1} \left\| \nabla_{\boldsymbol{w}}^{(t)} - \nabla_{\boldsymbol{w}^*}^{(t)} \right\|$$

$$= \frac{\eta}{2} \cdot \underset{\mathrm{Tr}_{\boldsymbol{w}}, \mathrm{Tr}_{\boldsymbol{w}^*}}{\mathbb{E}} \sum_{i=1}^{|\mathcal{D}_{\mathrm{ft}}|} \sum_{t=0}^{T-1} \left\| \nabla_{w_i}^{(t)} - \nabla_{w_i^*}^{(t)} \right\|. \tag{24}$$

Here, $\mathrm{Tr}_{\boldsymbol{w}}$ and $\mathrm{Tr}_{\boldsymbol{w}^*}$ denote the SGLD sampling trajectories drawn from the target distributions $p(\mathbf{w}, \boldsymbol{\theta} \mid \mathcal{D}_{\mathrm{ft}}, \mathcal{D}_{\mathrm{safe}})$ and $p(\mathbf{w}^*, \boldsymbol{\theta}^* \mid \mathcal{D}_{\mathrm{ft}}, \mathcal{D}_{\mathrm{safe}}, \mathcal{D}_{\mathrm{ft}}^{\mathrm{clean}})$, respectively. The norm used is the $\ell_1$-norm.

## I.4 Understanding Posterior Bias under Identity Weight Transformation

As shown in Eq. (24), the gradient term $\nabla_{\boldsymbol{w}}$ at each state contributes directly to the posterior bias. Here, we consider the identity weight transformation, which is defined as follows:

$$\sigma(w_i) = w_i. \tag{25}$$

Under identity weight transformation, the weight $\boldsymbol{w}$ updates in a monotonically decreasing manner:

$$\begin{cases} \underbrace{-\ell\left(\boldsymbol{z}_i^{\mathrm{ft}}; \boldsymbol{\theta}\right)}_{\nabla_{w_i}} \ll \underbrace{-\ell\left(\boldsymbol{z}_i^{\mathrm{ft}}; \boldsymbol{\theta}^*\right)}_{\nabla_{w_i^*}} \approx 0, \text{ if } \boldsymbol{z}_i^{\mathrm{ft}} \in \mathcal{D}_{\mathrm{ft}}^{\mathrm{benign}} \\ \underbrace{-\ell\left(\boldsymbol{z}_j^{\mathrm{ft}}; \boldsymbol{\theta}\right)}_{\nabla_{w_j}} \approx \underbrace{-\ell\left(\boldsymbol{z}_j^{\mathrm{ft}}; \boldsymbol{\theta}^*\right)}_{\nabla_{w_j^*}} \ll 0, \text{ if } \boldsymbol{z}_j^{\mathrm{ft}} \in \mathcal{D}_{\mathrm{ft}}^{\mathrm{harmful}} \end{cases} \tag{26}$$

The posterior bias under identity weight transformation arises mainly from updates in benign tuning data $\boldsymbol{z}_i^{\mathrm{ft}} \in \mathcal{D}_{\mathrm{ft}}^{\mathrm{benign}}$. Here, $\boldsymbol{\theta}^* \sim p(\boldsymbol{\theta}^* \mid \mathcal{D}_{\mathrm{ft}}, \mathcal{D}_{\mathrm{safe}}, \mathcal{D}_{\mathrm{ft}}^{\mathrm{val}})$ fits an additional dataset $\mathcal{D}_{\mathrm{ft}}^{\mathrm{val}}$, while $\boldsymbol{\theta} \sim p(\boldsymbol{\theta} \mid \mathcal{D}_{\mathrm{ft}}, \mathcal{D}_{\mathrm{safe}})$ does not. As a result, $\boldsymbol{\theta}^*$ achieves a much lower loss on benign data, such that $-\ell\left(\boldsymbol{z}_i^{\mathrm{ft}}; \boldsymbol{\theta}\right) \ll -\ell\left(\boldsymbol{z}_i^{\mathrm{ft}}; \boldsymbol{\theta}^*\right)$. Since the gradient of $\boldsymbol{w}$ is proportional to the negative loss, this results in a large and monotonically decreasing reduction in the weights for benign data, leading to poor fine-tuning performance. For harmful data $\boldsymbol{z}_j^{\mathrm{ft}} \in \mathcal{D}_{\mathrm{ft}}^{\mathrm{harmful}}$, which has different distribution from both $\mathcal{D}_{\mathrm{ft}}^{\mathrm{benign}}$ and $\mathcal{D}_{\mathrm{ft}}^{\mathrm{val}}$, the loss remains high and the weights also monotonically decrease.

## I.5 Theorem of Time-Weighted Accumulation of Posterior Bias

While BDS demonstrates superior defensive performance, a theoretical understanding of the inherent suboptimality in inferred data weights offers deeper insights that can guide future algorithmic improvements. Here, we further explore weight suboptimality under identity transformation through the theoretical perspective of posterior bias:

**Theorem I.4** (Time-Weighted Accumulation of Posterior Bias, identical to Theorem 4.2 in the main paper). *Let* $(\mathrm{Tr}_{\boldsymbol{w}}, \mathrm{Tr}_{\boldsymbol{\theta}})$ *and* $(\mathrm{Tr}_{\boldsymbol{w}^*}, \mathrm{Tr}_{\boldsymbol{\theta}^*})$ *denote the Stochastic Gradient Langevin Dynamics (SGLD) sampling trajectories drawn from the target distributions* $p(\boldsymbol{w}, \boldsymbol{\theta} \mid \mathcal{D}_{\mathrm{ft}}, \mathcal{D}_{\mathrm{safe}})$ *and* $p\left(\boldsymbol{w}^*, \boldsymbol{\theta}^* \mid \mathcal{D}_{\mathrm{ft}}, \mathcal{D}_{\mathrm{safe}}, \mathcal{D}_{\mathrm{ft}}^{\mathrm{val}}\right)$, *respectively. Here,* $\mathcal{D}_{\mathrm{ft}}^{\mathrm{val}}$ *is a held-out clean tuning dataset, serving as a test or validation set to evaluate fine-tuning performance. Then, the following proportionality holds:*

$$\underbrace{\underset{\mathrm{Tr}_{\boldsymbol{w}}, \mathrm{Tr}_{\boldsymbol{w}^*}}{\mathbb{E}} \left\| \boldsymbol{w}^{(T)} - \boldsymbol{w}^{*(T)} \right\|}_{\mathcal{PB}^{(T)}} \propto \underbrace{\underset{\mathrm{Tr}_{\bar{\boldsymbol{w}}}, \mathrm{Tr}_{\bar{\boldsymbol{w}}^*}}{\mathbb{E}} \sum_{t=0}^{T-1} (T-t) \left\| \boldsymbol{w}^{(t)} - \boldsymbol{w}^{*(t)} \right\|}_{\sum_{t=1}^{T-1} (T-t)\mathcal{PB}^{(t)}}. \tag{27}$$

*Here,* $\boldsymbol{w}^{(T)}$ *and* $\boldsymbol{w}^{*(T)}$ *represent the sampled weights at iteration* $T$, *while* $\bar{\boldsymbol{w}}^{(t)}$ *and* $\bar{\boldsymbol{w}}^{*(t)}$ *denote the intermediate states of the trajectories at iteration* $t$. *The term* $\mathcal{PB}^{(T)}$ *quantifies the posterior bias at the iteration* $T$, *and the summation* $\sum_{t=1}^{T-1} (T-t)\mathcal{PB}^{(t)}$ *characterizes the cumulative posterior bias over the entire sampling trajectory.*

**Implication.** Notably, the time-weighted factor $T - t$ emphasizes the greater influence of earlier iterations on the cumulative bias, highlighting the cumulative effect of initial steps on the posterior inference. This suggests that suboptimal sampling in the early stages of SGLD sampling trajectory can propagate and aggressively affect the overall posterior inference. Future work could focus on developing bias management strategies, particularly for mitigating bias in early stages of SGLD sampling trategories.

### I.6 Proof of theorem Theorem 4.2

*Proof.* Based on Definition I.3, the calculation formula for the SGLD-based posterior bias at $T$ iteration is given as follows:

$$\mathcal{PB}^{(T)} = \underset{\mathrm{Tr}_{\boldsymbol{w}}, \mathrm{Tr}_{\boldsymbol{w}^*}}{\mathbb{E}} \left\| \boldsymbol{w}^{(T)} - \boldsymbol{w}^{*(T)} \right\|$$

$$= \frac{\eta}{2} \cdot \underset{\mathrm{Tr}_{\boldsymbol{w}}, \mathrm{Tr}_{\boldsymbol{w}^*}}{\mathbb{E}} \sum_{t=0}^{T-1} \left\| \nabla_{\boldsymbol{w}}^{(t)} - \nabla_{\boldsymbol{w}^*}^{(t)} \right\| = \frac{\eta}{2} \cdot \underset{\mathrm{Tr}_{\boldsymbol{w}}, \mathrm{Tr}_{\boldsymbol{w}^*}}{\mathbb{E}} \sum_{i=1}^{|\mathcal{D}_{\mathrm{ft}}|} \sum_{t=0}^{T-1} \left\| \nabla_{w_i}^{(t)} - \nabla_{w_i^*}^{(t)} \right\|.$$

The SGLD sampling trajectory for $\boldsymbol{w}$ drawn from $p\left(\boldsymbol{w}, \boldsymbol{\theta} \mid \mathcal{D}_{\mathrm{ft}}, \mathcal{D}_{\mathrm{safe}}\right)$ is expressed as:

$$\mathrm{Tr}_{\boldsymbol{w}} = \boldsymbol{w}^{(0)} \to \cdots \to \boldsymbol{w}^{(T)}, \quad \boldsymbol{w}^{(t)} = \boldsymbol{w}^{(t)} + \frac{\eta}{2} \nabla_{\boldsymbol{w}}^{(t+1)} + \epsilon \sqrt{\eta},$$

$$\nabla_{w_i}^{(t)} = \nabla_{w_i} \log p(\boldsymbol{w}) - \ell(\boldsymbol{z}_i^{\mathrm{ft}}; \boldsymbol{\theta}^{(t)}).$$

Similarly, the SGLD sampling trajectory for $\boldsymbol{w}^*$ drawn from $p\left(\boldsymbol{w}^*, \boldsymbol{\theta}^* \mid \mathcal{D}_{\mathrm{ft}}, \mathcal{D}_{\mathrm{safe}}, \mathcal{D}_{\mathrm{ft}}^{\mathrm{val}}\right)$ is expressed as:

$$\mathrm{Tr}_{\boldsymbol{w}^*} = \boldsymbol{w}^{*(0)} \to \cdots \to \boldsymbol{w}^{*(T)}, \quad \boldsymbol{w}^{*(t+1)} = \boldsymbol{w}^{(t)} + \frac{\eta}{2} \nabla_{\boldsymbol{w}^*}^{(t)} + \epsilon \sqrt{\eta},$$

$$\nabla_{w_i^*}^{(t)} = \nabla_{w_i^*} \log p(\boldsymbol{w}) - \ell(\boldsymbol{z}_i^{\mathrm{ft}}; \boldsymbol{\theta}^{*(t)}).$$

Similarly, the SGLD sampling trajectory of $\boldsymbol{\theta}$ drawn from $p\left(\boldsymbol{w}, \boldsymbol{\theta} \mid \mathcal{D}_{\mathrm{ft}}, \mathcal{D}_{\mathrm{safe}}\right)$ is expressed as:

$$\mathrm{Tr}_{\boldsymbol{\theta}} = \boldsymbol{\theta}^{(0)} \to \cdots \to \boldsymbol{\theta}^{(T)}, \quad \boldsymbol{\theta}^{(t+1)} = \boldsymbol{\theta}^{(t)} + \frac{\eta}{2} \nabla_{\boldsymbol{\theta}}^{(t)} + \epsilon \sqrt{\eta},$$

$$\nabla_{\boldsymbol{\theta}}^{(t)} = \nabla_{\boldsymbol{\theta}} \log p(\boldsymbol{\theta}) - \sum_{i=1}^{|\mathcal{D}_{\mathrm{ft}}|} w_i^{(t)} \nabla_{\boldsymbol{\theta}} l\left(\boldsymbol{z}_i^{\mathrm{ft}}; \boldsymbol{\theta}^{(t)}\right).$$

Similarly, the SGLD sampling trajectory of $\boldsymbol{\theta}^*$ drawn from $p\left(\boldsymbol{w}^*, \boldsymbol{\theta}^* \mid \mathcal{D}_{\mathrm{ft}}, \mathcal{D}_{\mathrm{safe}}, \mathcal{D}_{\mathrm{ft}}^{\mathrm{val}}\right)$ is expressed as:

$$\mathrm{Tr}_{\boldsymbol{\theta}^*} = \boldsymbol{\theta}^{*(0)} \to \cdots \to \boldsymbol{\theta}^{*(T)}, \quad \boldsymbol{\theta}^{*(t+1)} = \boldsymbol{\theta}^{*(t)} + \frac{\eta}{2} \nabla_{\boldsymbol{\theta}}^{*(t)} + \epsilon \sqrt{\eta},$$

$$\nabla_{\boldsymbol{\theta}^*}^{(t)} = \nabla_{\boldsymbol{\theta}^*} \log p(\boldsymbol{\theta}^*) - \sum_{i=1}^{|\mathcal{D}_{\mathrm{ft}}|} w_i^{*(t)} \nabla_{\boldsymbol{\theta}^*} l\left(\boldsymbol{z}_i^{\mathrm{ft}}; \boldsymbol{\theta}^{*(t)}\right) - \sum_{j=1}^{|\mathcal{D}_{\mathrm{safe}}|} \nabla_{\boldsymbol{\theta}^*} l\left(\boldsymbol{z}_j^{\mathrm{safe}}; \boldsymbol{\theta}^{*(t)}\right).$$

Under the identity transformation, we compute the gradient difference for $w_i$ at the $t$-th iteration as follows:

$$||\nabla_{w_i}^{(t)} - \nabla_{w_i^*}^{(t)}|| = ||\ell(\boldsymbol{z}_i^{\mathrm{ft}}; \boldsymbol{\theta}^{*(t)}) - \ell(\boldsymbol{z}_i^{\mathrm{ft}}; \boldsymbol{\theta}^{(t)})|| = ||\Delta \ell_i^{(t)}||.$$

Here, the difference in loss is approximated using the Taylor expansion, yielding:

$$||\Delta \ell_i^{(t)}|| = ||\ell(\boldsymbol{z}_i^{\text{ft}}; \boldsymbol{\theta}^{*(t)}) - \ell(\boldsymbol{z}_i^{\text{ft}}; \boldsymbol{\theta}^{(t)})|| \approx \left\| \nabla_{\boldsymbol{\theta}} \ell \left( \boldsymbol{z}_i^{\text{ft}}; \boldsymbol{\theta}^{(t)} \right)^{\top} \left( \boldsymbol{\theta}^{*(t)} - \boldsymbol{\theta}^{(t)} \right) \right\|$$

$$= \frac{\eta}{2} \cdot \sum_{k=0}^{t-1} \left\| \underbrace{\nabla_{\boldsymbol{\theta}} \ell \left( \boldsymbol{z}_i^{\text{ft}}; \boldsymbol{\theta}^{(t)} \right)^{\top}}_{:= \nabla_{\boldsymbol{\theta}} \ell^{\top}} \left( \nabla_{\boldsymbol{\theta}^*}^{(k)} - \nabla_{\boldsymbol{\theta}}^{(k)} \right) \right\|$$

$$= \frac{\eta}{2} \cdot \sum_{k=0}^{t-1} \left\| \nabla_{\boldsymbol{\theta}} \ell^{\top} \left( \sum_{i=1}^{|\mathcal{D}_{\text{f}}|} \left( w_i^{*(k)} - w_i^{(k)} \right) \nabla_{\boldsymbol{\theta}} \ell \left( \boldsymbol{z}_i^{\text{ft}}; \boldsymbol{\theta}^{(k)} \right) + \sum_{j=1}^{|\mathcal{D}_{\text{safe}}|} \left[ \nabla_{\boldsymbol{\theta}} \ell \left( \boldsymbol{z}_j^{\text{safe}}; \boldsymbol{\theta}^{(k)} \right) - \nabla_{\boldsymbol{\theta}} \ell \left( \boldsymbol{z}_j^{\text{safe}}; \boldsymbol{\theta}^{*(k)} \right) \right] \right) \right\|$$

$$= \frac{\eta}{2} \cdot \sum_{k=0}^{t-1} \left\| \nabla_{\boldsymbol{\theta}} \ell^{\top} \left( \left( \boldsymbol{w}^{*(k)} - \boldsymbol{w}^{(k)} \right)^{\top} \nabla_{\boldsymbol{\theta}} \ell \left( \mathcal{D}^{\text{ft}}; \boldsymbol{\theta}^{(k)} \right) + \sum_{j=1}^{|\mathcal{D}_{\text{safe}}|} \left[ \nabla_{\boldsymbol{\theta}} \ell \left( \boldsymbol{z}_j^{\text{safe}}; \boldsymbol{\theta}^{(k)} \right) - \nabla_{\boldsymbol{\theta}} \ell \left( \boldsymbol{z}_j^{\text{safe}}; \boldsymbol{\theta}^{*(k)} \right) \right] \right) \right\|$$

$$\approx \frac{\eta}{2} \cdot \sum_{k=0}^{t-1} \left\| \nabla_{\boldsymbol{\theta}} \ell \left( \boldsymbol{z}_i^{\text{f}}; \boldsymbol{\theta}^{(t)} \right)^{\top} \left( \left( \boldsymbol{w}^{*(k)} - \boldsymbol{w}^{(k)} \right)^{\top} \nabla_{\boldsymbol{\theta}^{(k)}} \ell \left( \mathcal{D}^{\text{ft}}; \boldsymbol{\theta}^{(k)} \right) \right) \right\|$$

$$\propto \frac{\eta}{2} \cdot \sum_{k=0}^{t-1} \left\| \nabla_{\boldsymbol{\theta}} \ell \left( \boldsymbol{z}_i^{\text{ft}}; \boldsymbol{\theta}^{(t)} \right) \right\| \cdot \left\| \boldsymbol{w}^{*(k)} - \boldsymbol{w}^{(k)} \right\| \cdot \left\| \nabla_{\boldsymbol{\theta}^{(k)}} \ell \left( \mathcal{D}^{\text{ft}}; \boldsymbol{\theta}^{(k)} \right) \right\|.$$

The gradient term is calculated as follows:

$$\nabla_{\boldsymbol{\theta}} \ell \left( \mathcal{D}^{\text{ft}}; \boldsymbol{\theta}^{(k)} \right) = \begin{bmatrix} \nabla_{\boldsymbol{\theta}^{(k)}} l \left( \boldsymbol{z}_1^{\text{ft}}; \boldsymbol{\theta}^{(k)} \right) \\ \nabla_{\boldsymbol{\theta}^{(k)}} l \left( \boldsymbol{z}_2^{\text{ft}}; \boldsymbol{\theta}^{(k)} \right) \\ \vdots \\ \nabla_{\boldsymbol{\theta}^{(k)}} l \left( \boldsymbol{z}_{|\mathcal{D}_{\text{ft}}|}^{\text{ft}}; \boldsymbol{\theta}^{(k)} \right) \end{bmatrix}$$

The gradient difference is proportional to the cumulative weight difference over previous steps:

$$\|\nabla_{w_i}^{(t)} - \nabla_{w_i^*}^{(t)}\| \propto \sum_{k=0}^{t-1} \|\boldsymbol{w}^{*(k)} - \boldsymbol{w}^{(k)}\|.$$

We substitute this into the definition of $\mathcal{PB}^{(T)}$:

$$\mathcal{PB}^{(T)} \propto \mathbb{E} \sum_{t=0}^{T-1} \sum_{k=0}^{t-1} \|\boldsymbol{w}^{*(k)} - \boldsymbol{w}^{(k)}\|.$$

Rearrange the double summation by swapping the summation order:

$$\sum_{t=0}^{T-1} \sum_{k=0}^{t-1} = \sum_{k=0}^{T-1} \sum_{t=k+1}^{T}.$$

Substitute this back into the equation:

$$\mathcal{PB}^{(T)} \propto \mathbb{E} \sum_{k=0}^{T-1} (T-k) \|\boldsymbol{w}^{*(k)} - \boldsymbol{w}^{(k)}\|,$$

where $(T-k)$ represents the remaining time steps from step $k$ to $T$.

Transforming this with the expression of $\mathcal{PB}^{(t)}$, we obtain:

$$\mathcal{PB}^{(T)} \propto \sum_{t=0}^{T-1} (T-t) \mathcal{PB}^{(t)}.$$

$\square$

